# Optical properties and simple forcing efficiency of the organic aerosols and black carbon emitted by residential wood burning in rural Central Europe

Andrea Cuesta-Mosquera[1], Kristina Glojek[2], Griša Močnik[2,3,4], Luka Drinovec[2,3,4], Asta Gregorič[2,5], Martin Rigler[5], Matej Ogrin[6], Baseerat Romshoo[1], Kay Weinhold[1], Maik Merkel[1], Dominik van Pinxteren[7], Hartmut Herrmann[7], Alfred Wiedensohler[1], Mira Pöhlker[1], Thomas Müller[1]

[1]Department of Experimental Aerosol and Cloud Microphysics, Leibniz Institute for Tropospheric Research (TROPOS), Leipzig, 04318, Germany
[2]Center for Atmospheric Research, University of Nova Gorica, Ajdovščina, 5270, Slovenia
[3]Department of Condensed Matter Physics, Jožef Stefan Institute, Ljubljana, 1000, Slovenia
[4]Haze Instruments d.o.o., Ljubljana, 1000, Slovenia
[5]Aerosol d.o.o., Ljubljana, 1000, Slovenia
[6]Department of Geography, Faculty of Arts, University of Ljubljana, Ljubljana, 1000, Slovenia
[7]Atmospheric Chemistry Department, Leibniz Institute for Tropospheric Research (TROPOS), Leipzig, 04318, Germany

*Correspondence to*: Andrea Cuesta-Mosquera (cuesta@tropos.de, andrea.cuesta2305@gmail.com)

**Abstract.** Recent years have seen an increase in the use of wood for energy production of over 30 %, and this trend is expected to continue due to the current energy crisis and geopolitical instability. At present, residential wood burning (RWB) is one of the most important sources of organic aerosols (OA) and black carbon (BC), posing a significant risk to air quality and health. Simultaneously, as a substantial aerosol source, RWB also holds relevance in the context of aerosol radiative effects and climate. While BC is recognized for its large light absorption cross-section, the role of OA in light absorption is still under evaluation due to their heterogeneous composition and source-dependent optical properties. Existing studies that characterize wood-burning aerosol emissions in Europe primarily concentrate on urban and background sites and focus on BC properties. Despite the significant RWB emissions in rural areas, these locations have received comparatively less attention. The present scenario underscores the imperative for an improved understanding of RWB pollution, aerosol optical properties, and their subsequent connection to climate impacts, particularly in rural areas.

We have characterized atmospheric aerosol particles from a central European rural site during wintertime in the village of Retje in Loški Potok, Slovenia, from 01.12.2017 to 07.03.2018. The village experienced extremely high aerosol concentrations produced by RWB and near-ground temperature inversion. The isolated location of the site and the substantial local emissions made it an ideal laboratory-like place for characterizing RWB aerosols with low influence from non-RWB sources under ambient conditions. The mean mass concentrations of OA and BC were 35 µg m$^{-3}$ (max = 270 µg m$^{-3}$) and 3.1 µg m$^{-3}$ (max = 24 µg m$^{-3}$), respectively. The mean total particle number concentration (10–600 nm) was 9.9 x 10$^3$ particles cm$^{-3}$ (max = 59 x 10$^3$ particles cm$^{-3}$). The mean total light absorption coefficient at 370 nm and 880 nm measured by an Aethalometer AE33 were 120 Mm$^{-1}$ and 22 Mm$^{-1}$ and had maximum values of 1100 Mm$^{-1}$ and 180 Mm$^{-1}$, respectively. The aerosol concentrations

and absorption coefficients measured during the campaign in Loški Potok were significantly larger than those reported values for several urban areas in the region with larger populations and extent of aerosol sources.

Here, considerable contributions from brown carbon (BrC) to the total light absorption were identified, reaching up to 60 % and 48 % in the near UV (370 nm) and blue (470 nm) wavelengths. These contributions are up to three times higher than values reported for other sites impacted by wood-burning emissions. The calculated mass absorption cross-section and the absorption Ångström exponent for RWB OA were $MAC_{OA, 370 nm}= 2.4$ $m^2$ $g^{-1}$, and $AAE_{BrC, 370-590 nm}= 3.9$, respectively.

Simple forcing efficiency (SFE) calculations were performed as a sensitivity analysis to evaluate the climate impact of the RWB aerosols produced at the study site by integrating the optical properties measured during the campaign. The SFE results show a considerable forcing capacity from the local RWB aerosols, with a high sensitivity to OA absorption properties and a more substantial impact over bright surfaces like snow, typical during the coldest season with higher OA emissions from RWB. Our study's results are highly significant regarding air pollution, optical properties, and climate impact. The findings suggest that there may be an underestimation of RWB emissions in rural Europe and that further investigation is necessary.

## 1 Introduction

Burning woody biomass for heating and cooking has progressively increased in recent decades. Between 2009 and 2015, this increment was 34 % in Europe (Camia et al., 2021). Government incentives to use "renewable" energy sources to cut fossil fuel dependence and reduce greenhouse gas emissions have contributed to this rise. In addition, the use of residential wood burning (RWB) as an energy source is significantly growing in urban and rural areas due to higher energy costs and uncertainties from the current geopolitical instability. In the winter of 2023-2024, the increasing cost of natural gas is expected to boost RWB in the region, where private households account for more than 40% of the total wood consumption for energy use (UNECE, 2022).

RWB emissions significantly contribute to ultrafine (UFP, < 100 nm) and fine (< 2.5 μm) aerosol particle emissions (Casquero-Vera et al., 2021; Ozgen et al., 2017). The composition of these aerosol particles includes substantial fractions of organic aerosols (OA) and black carbon (BC) (Fine et al., 2001; Liang et al., 2021). During winter, biomass burning is the principal source of primary OA emissions in the region, and, together with solid biofuels and coal combustion, it is also a primary source of BC (European Environmental Agency, 2020; Herich et al., 2014). Global emission inventories indicate that residential biomass burning produces 35 % of BC emissions, occupying an important position above other emission sources, such as on-road diesel vehicle emissions (26 %) (Xu et al., 2021).

The above figures are of major concern due to the adverse effects of high-content OA and BC particles on human health and climate. Black carbon has been linked to increased morbidity and mortality (Geng et al., 2013) since it carries toxic substances affecting the immune and respiratory systems as well as cardiac function (Janssen et al., 2011). Fine and ultrafine aerosol particles are potentially harmful because their size allows them to penetrate deeper into the respiratory tract and are associated with proinflammatory effects on human cells (Corsini et al., 2017; WHO et al., 2012). Organic aerosols have been connected

to cardiovascular and respiratory affectations (Mauderly and Chow, 2008). The OA heterogeneous composition includes Polycyclic Aromatic Hydrocarbons (PAHs), some of which have carcinogenic, mutagenic, and genotoxic properties (Chowdhury et al., 2022; Vicente and Alves, 2018, and references therein).

On the other hand, OA and BC have direct and indirect impacts on climate. Black carbon particles alter the Earth's radiative balance due to their capacity to absorb solar radiation, leading to a direct warming effect on the atmosphere (Bond et al., 2013). Indirect effects include altering lifetime and cloud formation processes since the aerosol particles can act as cloud condensation nuclei and alter the cloud mixing state (Chen et al., 2018; Koch et al., 2011). In the case of OA, these were historically considered as scattering compounds responsible for an atmospheric cooling effect; nevertheless, more recent studies evidence the warming capacity from the absorbing carbonaceous fraction present in the OA, named brown carbon (BrC) (Laskin et al., 2015, and references therein). Often, organic aerosols coating BC particles enhance the particle light absorption by the called "lensing effect," which consists of the concentration of photons in the particle BC core due to the presence of scattering species in the particle coating; this enhancement varies along the visible and infrared spectrum (Bond et al., 2013; Kalbermatter et al., 2022; Zhang et al., 2018). Furthermore, BrC absorbs light in the shorter UV and visible wavelengths, and its inclusion in climate models has shown that the direct OA radiative forcing at the top of the atmosphere turns from cooling (negative) to warming (positive) (Feng et al., 2013).

The OA contribution to atmospheric warming processes is still being discussed. The aerosol light absorption attributed to OA (BrC) and its radiative forcing effects vary regionally depending on the emission source and the atmospheric and burning conditions. In an urban Indian site, Shamjad et al. (2016) found that BrC emissions contributed 29 % of the total light absorption at 405 nm; in multiple locations all over France, Zhang et al. (2020) found BrC contributions ranging from 18 to 42 % at 370 nm; for eastern China, Wang et al. (2018) reported a range of 10 to 24 % (33 % maximum) as the contribution of BrC to the total aerosol light absorption at 370 nm. The regional variability in the OA optical properties represents a challenge for global climate models and makes the constraint of the net global warming or cooling effects difficult from residential burning emissions (Kodros et al., 2015; Szopa et al., 2021). These facts show the need for source-oriented and highly temporally resolved field studies, especially in poorly characterized rural areas with significant RWB emissions, whose impacts can be underestimated (Vicente and Alves, 2018; Zhang et al., 2020).

In this work, the authors present the results of a field campaign performed in a European rural site with substantial RWB emissions. We evaluated the optical properties (light absorption coefficients ($b_{abs}$), mass absorption cross-section (MAC), and absorption Ångström exponent (AAE)) of the aerosol particles produced by a predominant aerosol source in an isolated village. The atmospheric stability during wintertime and the topography of the area favor the accumulation of locally produced pollutants and reduce the influence of external aerosol sources. Such circumstances allowed us to characterize aerosols produced by RWB under real conditions. Ultimately, we present the results from simple forcing efficiency estimations for the strongly light-absorbing BrC produced by RWB in the study site. We incorporated the measured optical properties in the simple forcing efficiency (SFE) calculations and compared the results by considering the OA as absorbing and non-absorbing species in the net radiative forcing (warming vs. cooling effect) over two types of surfaces (snow and Earth-average).

## 2 Methods

### 2.1 Study site and monitoring campaign

The field study was performed in the municipality of Loški Potok, Slovenia, in the rural village of Retje. The area is a small and shallow depression (area = 1.5 km$^2$, depth = 150 m, 45°42'34.8" N, 14°34'53.8" E, Fig. 1). The basin-shaped topography of the site favors the formation of near-ground temperature inversions, especially in the coldest season, contributing to the accumulation of atmospheric aerosols emitted locally (Glojek et al., 2022). During temperature inversions, the denser and cooler air remains at the bottom of the depression, stopping the mixing with warmer air at the top and inhibiting the vertical dilution of air pollutants.

The small village, with 690 inhabitants and 243 households, is encircled by a densely forested area; the local families use wood as the primary energy source for residential heating and cooking in the coldest season. The influence of aerosol traffic emissions is significantly low on the site, with an average of less than 100 vehicles circulating daily. The impact of industrial emissions is negligible since the nearest atmospheric industrial emission sources are more than 10 km from the monitoring site (Glojek et al., 2020, 2022).

The local topography, the atmospheric stability, and the distinctive and predominant source of aerosol emissions make Retje a unique field laboratory to investigate the optical properties of organic aerosols and black carbon produced by residential wood burning under realistic conditions. The campaign was performed during wintertime from 01 December 2017 to 07 March 2018. We used a "twin monitoring scheme" with two fixed monitoring stations operating simultaneously (Fig. 1 and Table 1). One station was located at the bottom of the hollow in the middle of the village (main station, 715 m a.s.l.), while the second station was located at Tabor hill (815 m a.s.l.) and served as a background station.

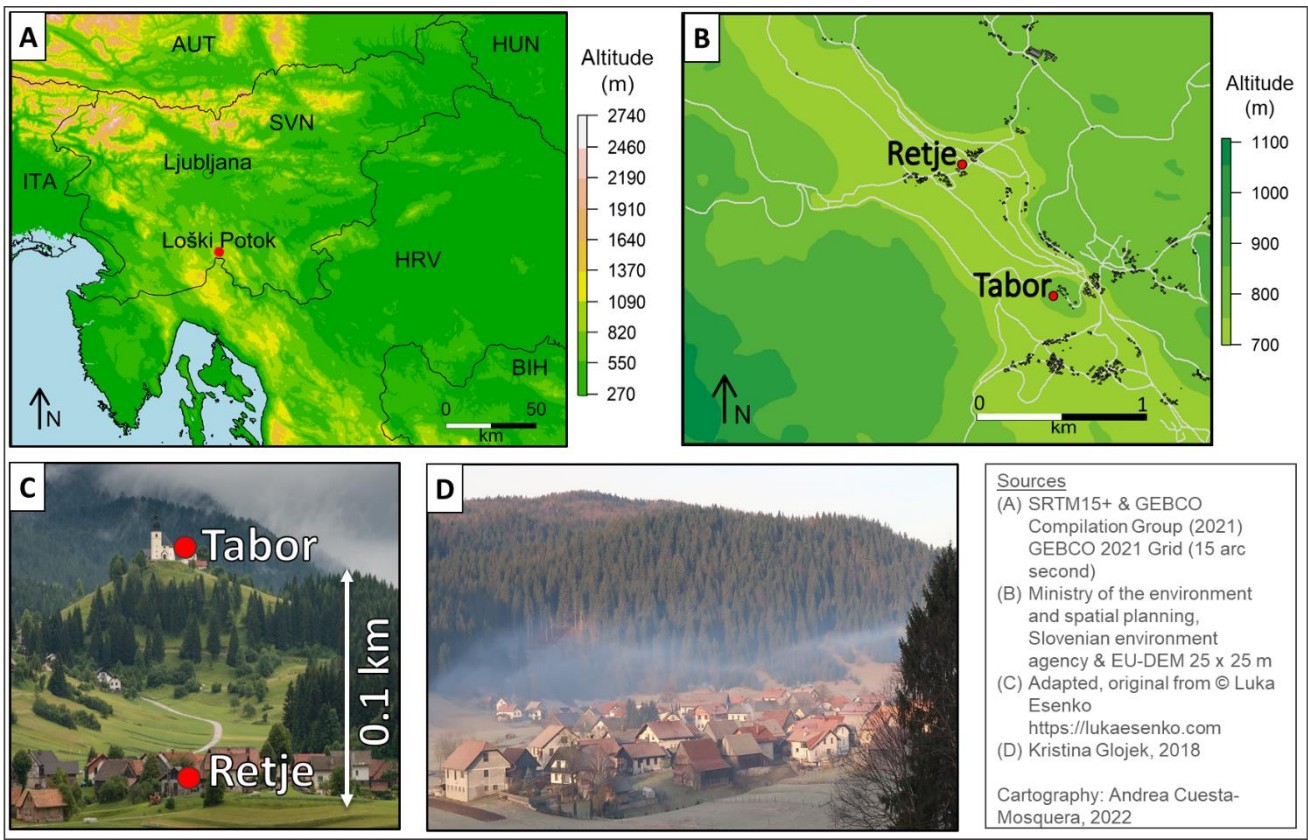

**Figure 1: Study site and the surrounding environment: (A) geographical location in Europe, (B) geographical location of the monitoring stations in Loški Potok, (C) photo of the valley and the monitoring stations, (D) photo of Retje during a near-ground temperature inversion (27 January 2018).**

## 2.2 Instrumentation

In both monitoring stations, the optical attenuation coefficient at seven wavelengths was continuously measured using filter-based absorption photometers (Aethalometers AE33, 370 to 950 nm, Aerosol Magee Scientific). The AE33s used filter tapes model M8060, made of PET polymerized polyester (59 %) and fiberglass (41 %). The particle number concentration (N) and particle number size distributions (PNSD) were monitored at Retje (10-800 nm) and Tabor hill (10-600 nm) using Mobility Particle Size Spectrometers (MPSS). At the village, a reference MPSS (TROPOS Ref. No. 1, high-voltage supply of positive

polarity) set up with a bipolar diffusion charger (radioactive source: 85 Kr), a Differential Mobility Particle Sizer (DMA, Hauke-type medium), and a butanol-type condensation particle counter (CPC, TSI model 3772) operated. Whereas, at the background station, a TSI MPSS (high-voltage supply of negative polarity) set up with a bipolar diffusion charger (radioactive source: x-ray), a DMA (TSI 3081), and a water-type CPC (TSI model 3785) was installed. A multiple-charge inversion routine was applied to the raw mobility distributions to calculate the final PNSD (Pfeifer et al., 2014); additional corrections were

applied to the PNSD, including CPC counting efficiency and particle losses due to diffusion in the inlet and sampling tubes and internal particle diffusion (corrected via the "equivalent pipe length" method, Wiedensohler et al., 2012).

Filter samples of particulate matter ($PM_{10}$) were collected every 12 hours at the village station (06:00 to 18:00 and 18:00 to 06:00, local time) using a high-volume sampler (DHA-80, Digitel). $PM_{10}$ mass concentrations were determined gravimetrically in the laboratory following the European Union standard EN 12341. The quartz fiber filters (150 mm diameter) were preheated before sampling for at least 24 hours at 105 °C to minimize blank values and frozen after sampling until their characterization in the laboratory; the weighing was done using a microbalance (AT261 Delta Range, Mettler-Toledo). The $PM_{10}$ filters were analyzed to estimate the particle's chemical composition, including organic and elemental carbon (OC/EC), ions ($NH_4^+$, $Cl^-$, $Na^+$, $K^+$, $Mg^{2+}$, $Ca^{2+}$, $NO_3^-$, $SO_4^{2-}$, and $C_2O_4^{2-}$), and levoglucosan. Mass concentrations of OC/EC were quantified following the EUSAAR-2 Protocol (Cavalli et al., 2010); ions were determined using ion chromatography (Dionex ICS3000) of ultrapure water extracts (further details in Fomba et al., 2014); levoglucosan was quantified using high-performance anion exchange chromatography coupled with an electrochemical detector (HPAEC-PAD, Iinuma et al., 2009).

The total carbon (TC) mass concentration at the village station was measured using an online Total Carbon Analyzer (TCA08, Aerosol Magee Scientific, Rigler et al., 2020). The dual-chamber instrument uses an online thermal method to quantify the total carbonaceous fraction in atmospheric aerosols. In one chamber, the sample is collected over a quartz fiber filter (47 mm diameter) and heated to 940 °C, transforming the carbon compounds into $CO_2$. The amount of $CO_2$ is measured before and after combustion by a $CO_2$ detector and later integrated to calculate the total carbon mass concentration. Simultaneously, the second chamber collects a new aerosol sample (sampling time adjustable from 20 min to 24 h). Both chambers alternate between sampling and analysis, enabling online functionality. Further details about the MPSS, DHA-80, and TCA operating principles are given in the Supplement.

Cyclones with $PM_{2.5}$ cut-off were used to sample aerosols for the AE33, MPSSs, and TCA; Nafion®Permapure air dryers (length=1.5 m) kept the relative humidity of the AE33 and MPSS samples below 40 %. The instruments operated under ambient room temperature. The DHA-80 high-volume sampler was equipped with a $PM_{10}$ impactor-type inlet. Table 1 shows the specifications and operating conditions of the instruments used during the campaign.

The quality of the instrument measurements was assured by intercomparison, calibration, and maintenance in the laboratory. The TROPOS Ref. No. 1 MPSS and the TSI MPSS were calibrated and intercompared against a reference MPSS (TROPOS Reference MPSS No.4, set up with a CPC TSI model 3772) and a reference CPC (TSI CPC, model 3010) at the World Calibration Center for Aerosol Physics (WCCAP) in Germany. The comparison against the reference MPSS and CPC showed acceptable deviations within the range of ±10 % for the mean particle size distributions and the integrated number of particle concentrations). Further detailed results from the intercomparisons are shown in (Glojek et al., 2020, 2022). The AE33 and the TCA were maintained as the manufacturer indicated (Aerosol Magee Scientific, 2018, 2022), including verification and inspection of the inlet flow and sampling lines, flow verification and calibration, and leakage tests.

**The Aethalometer AE33 and multiple scattering harmonization**

The Aethalometer AE33 collects aerosol particles in a filter matrix with a high time resolution (1 sec or 1 min). The instrument

measures the change in the light transmitted through the filter loaded with aerosols (light attenuation) due to light absorption and scattering from the particles and the filter material. The AE33 creates two sample spots on the filter, which allows the online correction of the so-called *filter-loading effect*, an artifact caused by the reduced sensitivity to detect changes in the light attenuation produced by the accumulation of particles during sampling. More details about the AE33 algorithm and the loading compensation can be found in Drinovec et al. (2015) and the Supplement.

The internal algorithm of the AE33 converts the attenuation ($b_{ATN}(\lambda)$) into a light absorption coefficient ($b_{abs}(\lambda)$) using a correction factor to account for the multiple scattering of light ($C$ factor, see Eq. S1). The value of $C$ is set manually in the instrument and, according to the manufacturer, depends on the filter model used during the measurements. For instance, the newest filter tape (M8060) has a corresponding $C = 1.39$. Nevertheless, multiple studies point out that the scattering of light in filter-based absorption photometers is also affected by the aerosol particles deposited on the filter matrix according to their

single scattering albedo, and the scattering within the filter fibers (Ajtai et al., 2019; Bernardoni et al., 2021; Collaud Coen et al., 2010; Drinovec et al., 2022; Saturno et al., 2017; Yus-Díez et al., 2021). Furthermore, some studies suggest that condensation of semi-volatile organic compounds on the filter might contribute to an apparent absorption enhancement (Cappa et al., 2008; Weingartner et al., 2003).

The use of unrepresentative scattering correction factors might drive to an overestimation of the aerosol absorption in the AE33

and consequently, BC mass concentrations. Therefore, the values of $b_{abs}(\lambda)$ obtained from the instrument must be corrected when using the manufacturer $C$ alone. In this regard, the Aerosol, Clouds and Trace Gases Research Infrastructure (ACTRIS) proposes to include a scaling or harmonization factor to account for the aerosol effect in multiple light scattering. This "harmonization" factor attempts to harmonize the AE33 measurements with the MAAP measurements, which is considered the closest to a reference for online aerosol light absorption measurements. ACTRIS has calculated a median harmonization

factor ($H$) of 1.76 for AE33 measurements using the filter tape M8060 ($C = 1.39$), based on simultaneous AE33-MAAP measurements from multiple European sites (Müller and Fiebig, 2021). In the present study, we have used a harmonization factor to scale the light absorption coefficients from Loški Potok. Given the absence of simultaneous AE33-MAAP measurements in the study site, we used a value of $H = 1.9$ calculated from measurements performed by TROPOS at the village of Melpitz during winter (2018-2019). Melpitz is a small village (~200 inhabitants) located in eastern Germany (at 50 km from

Leipzig) where RWB is the predominant heating source during the coldest period (van Pinxteren et al., 2023). For Loški Potok, the light absorption coefficients $b_{abs}(\lambda)^{Harm.}$ were harmonized as follows:

$$b_{abs}(\lambda)^{Harm.} = \frac{b_{ATN}(\lambda)}{C*H} = \frac{b_{abs}(\lambda)}{H} \tag{1}$$

Note that the harmonized absorption coefficients will be referred to as $b_{abs}(\lambda)$ in the following sections.

**Table 1. Instrumentation and operating conditions.**

| Measurement | Instrument and manufacturer | Operating conditions | |
|---|---|---|---|
| | | **Time resolution** | **Configuration** |
| **Village (foreground) station** | | | |
| Particle number size distribution (particles cm$^{-3}$) | Mobility particle size spectrometer MPSS, TROPOS Ref. No. 1 (Hauke-type medium DMA) with a TSI CPC (condensation particle counter) model 3772 | 5 min | Sheath air to aerosol flow ratio: 5:1 L min$^{-1}$<br>Radioactive source: $^{85}$Kr<br>Mobility diameter range: 10-800 nm<br>Inlet: PM$_{2.5}$ |
| Aerosol light absorption (Mm$^{-1}$) | Aethalometer, model AE33, Aerosol Magee Scientific | 1 min | Air flow rate: 5 L min$^{-1}$<br>Filter tape: M8060<br>$C_{M8060} = 1.39$, harmonization factor = 1.9<br>λ: 370, 470, 520, 590, 660, 880 & 950 nm<br>Inlet: PM$_{2.5}$ |
| PM$_{10}$ mass concentration (μg m$^{-3}$) | High-volume PM sampler, model DHA-80, DIGITEL | 12 h | Air flow rate: 500 L min$^{-1}$ |
| Total carbon mass concentration (μg m$^{-3}$) | Total Carbon Analyzer, model TCA08, Aerosol Magee Scientific | 1 h | Air flow rate:<br>16.7 L min$^{-1}$ (Sample airflow), 0.5 L min$^{-1}$ (Analytic airflow)<br>Inlet: PM$_{2.5}$ |
| Temperature (°C), relative humidity (%), atmospheric pressure (mbar) | Meteorological sensor, TPR 159, AMES | 1 min | -- |
| **Tabor (background) station** | | | |
| Particle number size distribution (particles cm$^{-3}$) | Mobility particle size spectrometer MPSS, TSI Inc. (DMA model 3081) with a TSI CPC model 3785 | 5 min | Sheath air to aerosol flow ratio: 4.1:1 L min$^{-1}$<br>Radioactive source: x-ray<br>Mobility diameter range: 10-600 nm<br>Inlet: PM$_{2.5}$ |

| | | | Air flow rate: 5 L min |
| --- | --- | --- | --- |
| Aerosol light absorption (Mm$^{-1}$) | Aethalometer, model AE33, Aerosol Magee Scientific | 1 min | Filter tape: M8060<br>$C_{M8060} = 1.39$, harmonization factor = 1.9<br>λ: 370, 470, 520, 590, 660, 880 & 950 nm<br>Inlet: PM$_{2.5}$ |
| Temperature (°C), relative humidity (%), atmospheric pressure (mbar) | Meteorological sensor, TPR 159, AMES | 1 min | -- |

## 2.3 Hourly OA and BC mass concentrations

The PM$_1$ mass concentration was defined as the sum of BC, OA, and inorganic aerosols (InA) masses, similar to Setyan et al. (2012). Accordingly, the hourly OA mass concentration was calculated using a mass balance as follows:

$$[OA] = [PM_1]_{MPSS} - [eBC]_{AE33} - [InA]_{PM}, \tag{2}$$

For the previous calculation, we considered the good agreement between the PM$_{10}$ mass concentrations from the filters and the 12-hour averages of PM$_1$ from the MPSS (see Fig. 2, orthogonal fit: slope = 0.9, R$^2$ = 0.96). The comparison of PM$_1$-PM$_{10}$ suggests that most PM$_{10}$ is composed of PM$_1$ (~90 %). This approach, however, could be an important source of uncertainty in determining the hourly OA mass concentration since a fraction of OA might fall in the 1-2.5 microns size range; in contrast, the fraction of BC is most probably below the PM$_1$ fraction (section 3.3 shows an analysis of uncertainty).

The hourly PM$_1$ mass concentration was estimated from the MPSS volume size distribution, with the following assumptions: (i) the bulk aerosol density is constant (ρ = 1.4 g cm$^{-3}$, a typical value attributed to aerosols with relatively larger fractions of organic species (Rissler et al., 2014; Turpin and Lim, 2001)), and (ii) the aerosol particles have spherical shape (the mobility diameter is equivalent to the volume-equivalent diameter). Please note that the whole size distributions from the MPSS in Retje (mobility diameters from 10 to ~800 nm) were used in the PM$_1$ mass calculation to cover the aerodynamic diameter of

PM$_1$. Mass concentrations of PM$_1$ were not calculated for the background station, considering the maximum mobility diameter reached by the TSI MPSS.

The BC mass concentration was estimated from the optical measurement of absorption from the Aethalometer AE33, denoted consequently as equivalent black carbon (eBC) (Petzold et al., 2013). The local mass absorption cross-section for BC (MAC$_{BC}$) was determined as the slope from the orthogonal fit between the 12-hour averaged aerosol light absorption at 950 nm from the

AE33 b$_{abs}$(λ) and the EC mass concentration derived from the thermo-optical analysis of the PM$_{10}$ filters (see section 3.2.3).

Despite the differences between BC and EC given by their operational definitions (Bond and Bergstrom, 2006; Petzold et al., 2013), we assume, for this study, that both species are comparable and that at 950 nm BC dominates the light absorption (Zanatta et al., 2016). Accordingly, the eBC mass was calculated as follows:

$$eBC = \frac{b_{abs}(950\,nm)^{Harm.}}{MAC_{BC}(950\,nm)},$$ (3)

The mass fraction of InA in $PM_{10}$ ($Cl^-$, $NH_4^+$, $NO_3^-$, $SO_4^{2-}$, $C_2O_4^{2-}$, Na, K, Mg, and Ca) was, on average, 15 % of the total PM mass. For $PM_1$, we assumed a similar percentual contribution of InA, considering the correlation between $PM_{10}$ and $PM_1$ (see Fig. 2) and a homogeneous distribution of the inorganic aerosols in the PM mass. The uncertainty of the slope in Fig. 2 was

estimated using error propagation as outlined in section 3.3. This estimation accounted for the individual contributions of the errors involved in both the $PM_1$ calculation and the gravimetric determination of $PM_{10}$. The final uncertainty was determined to be 15%.

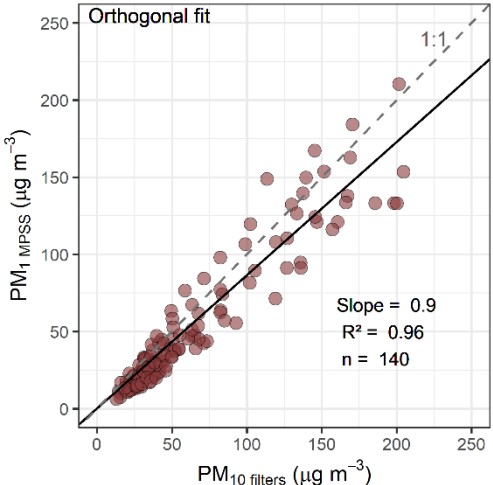

**Figure 2: Scatter plot and orthogonal regression (solid black line) for the PM₁ calculated from the MPSS and the PM₁₀ mass from**
**filters.** The figure includes the regression slope, the coefficient of determination (R²), and the number of observations (n). The intercept was forced through zero.

The estimated hourly OA mass concentration ($OA_{MPSS}$, Eq. 2) was compared to the OC mass from the Total Carbon Analyzer ($OC_{TCA}$, see Eq. S4 in the Supplement) and the OC from filters ($OC_{filters}$). In both cases, a good agreement was observed between OA and OC (Fig S1, $OA_{MPSS}$ vs. $OC_{TCA}$: $R^2 = 0.95$, $OA_{MPSS}$ vs. $OC_{filters}$: $R^2 = 0.97$), suggesting that $OA_{MPSS}$ is a good

approximation in the study site. The slopes in both comparisons ($OA_{MPSS}$ vs. $OC_{TCA}$: 2.0, $OA_{MPSS}$ vs. $OC_{filters}$: 1.6) represent the local OA/OC ratio. The estimated OA/OC differ due to the distinct analytical procedures from both methods and sampling periods (1-hour averaged data from TCA and 12-hour data from filters) (Brown et al., 2013). Nevertheless, both OA/OC fall within the range of ratios reported in the literature (Srinivas and Sarin, 2014b; Xing et al., 2013).

Please note that in the following, OA$_{MPSS}$ is referred to as OA.

## 2.4 Simple radiative forcing efficiency

To estimate the climate impact of the RWB aerosol particles emitted in Retje, we used the simple forcing efficiency (SFE), which quantifies the perturbation of the emitted aerosol particles on the radiative balance of the Earth's atmosphere (Chen and Bond, 2010; Choudhary et al., 2021). For this purpose, the equation proposed by Chen and Bond (2010) was used:

$$\frac{d\text{SFE}}{d\lambda} = -\frac{1}{4}\frac{dS(\lambda)}{d\lambda}\tau_{atm}^2(1 - F_c)[2(1 - a_s)^2 * \beta(\lambda) * \text{MSC}(\lambda) - 4a_s * \text{MAC}(\lambda)], \tag{4}$$

Where SFE is given in W g$^{-1}$, dS($\lambda$)/d$\lambda$ is the solar irradiance (W m$^{-2}$ nm$^{-1}$) obtained from the reference solar spectra from ASTM G173-03, $\tau_{atm}$ and $a_s$ are the atmospheric transmission (0.79) and the surface albedo (Earth average = 0.19, fresh snow = 0.80), respectively, whose values were taken from the study of Chen and Bond (2010). F$_c$ is the cloud fraction (assumed 0.6), $\beta(\lambda)$ is the light backscatter fraction, and MAC($\lambda$) and MSC($\lambda$) are the mass absorption and mass scattering cross-sections (m$^2$ g$^{-1}$), respectively. $\beta(\lambda)$ was estimated by the mathematical relation with the asymmetry parameter $g(\lambda)$ proposed by Sagan and Pollack (1967):

$$\beta(\lambda) = \frac{1}{2}(1 - g(\lambda)), \tag{5}$$

## Mie modeling

The $g(\lambda)$ and MSC($\lambda$) were estimated using the core-shell Mie theory (Bohren and Huffman, 1998). The aerosol particles are considered to be spherical, consisting of a BC core surrounded by OA and InA, which are homogeneously mixed in the shell. As input for the Mie modeling, we provided volumetric fractions, complex refractive indexes, and size distributions of the three components every 12 hours. The volumetric fractions were calculated considering the mass concentrations of OA, EC, and InA from the PM$_{10}$ filters and densities from the literature (Eq. S6-S8): $\rho_{BC}$ = 1.8 g cm$^{-3}$, $\rho_{OA}$ = 1.4 g cm$^{-3}$, and $\rho_{InA}$ = 2.1 g cm$^{-3}$ (Bond and Bergstrom, 2006; Li et al., 2016; Turpin and Lim, 2001). The wavelength-dependent complex refractive indexes (n -real part and k -imaginary part) of BC, OA, and InA were also taken from previous studies and are shown in Table S1 (Chen and Bond, 2010; Kim et al., 2015). Finally, the effective complex refractive index of the shell was calculated using a volume mixing rule (Eq. S9). The MAC($\lambda$) for the simulated core-shell particles were calculated using field measurements of particle mass and absorption, as follows:

$$\text{MAC}(\lambda) = \frac{b_{abs}(\lambda)}{PM_1}, \tag{6}$$

## 2.5 Atmospheric stability

The different categories of atmospheric stability in Loški Potok were identified according to the potential temperature gradient

$(\partial\theta/\partial z)$, expressed in K m$^{-1}$ and calculated as the difference of the absolute potential temperature ($\theta$) from both sites (Eq. 7): Tabor hill at the top of the valley ($\theta_{background}$) and the Retje village at the bottom ($\theta_{village}$). Using the potential temperature is advantageous since it discounts the compressibility effect caused by the air pressure. $\theta$ is the temperature a parcel of air would have if it expands or compresses adiabatically (no heat is added or subtracted) and is brought to a reference pressure (Hartmann, 2016). Hourly values of $\theta$ for Retje and Tabor were calculated as shown in Eq. 8, using the measurements from

the meteorological sensors operating during the campaign (Table 1).

$$\frac{\partial\theta}{\partial z} = (\theta_{background} - \theta_{village}) / dz \tag{7}$$

$$\theta = T * \left(\frac{P_0}{P}\right)^{R/Cp}, \tag{8}$$


where T is the absolute temperature, $P_0$ is the reference pressure (standard atmospheric pressure, 1 atm), P is the local pressure, R is the gas constant for air, and Cp is the specific heat capacity of air at constant pressure ($R/Cp = 0.286$ for ambient air (Wallace and Hobbs, 2006)).

The categories of atmospheric stability are given according to $\partial\theta/\partial z$: When $\partial\theta/\partial z > 0$, the potential temperature increases

with height, the atmosphere is stable, and near-ground temperature inversions occur; when $\partial\theta/\partial z = 0$, the potential temperature is uniform, there are no net upward or downward buoyancy forces and the atmosphere is neutral; when $\partial\theta/\partial z < 0$, there is a decrease in the potential temperature, the atmosphere is unstable and vertical motions within an air parcel are favored (Hartmann, 2016). During the unstable conditions at Loški Potok, $\partial\theta/\partial z$ ranged from -5 K m$^{-1}$ to 0 K m$^{-1}$, while for stable conditions $\partial\theta/\partial z$ ranged from 0 K m$^{-1}$ to 12 K m$^{-1}$. The wide range of positive $\partial\theta/\partial z$ motivated us to partition the

inversion periods into two subcategories: weak inversion and strong inversion. The final categories of atmospheric stability used in the analysis are:

**Table 2. Categories of atmospheric stability according to the potential temperature gradient.**

| Atmospheric stability | Potential temperature gradient (K m$^{-1}$)* |
|---|---|
| Weak inversion | $0 < \partial\theta/\partial z < 2$ |
| Strong inversion | $\partial\theta/\partial z > 2$ |
| Unstable | $\partial\theta/\partial z < 0$ |
| Neutral | $\partial\theta/\partial z = 0$ |

*Figure 3d shows the hourly potential temperature gradients ($\partial\theta/\partial z$) and the respective categories of atmospheric stability at Loški Potok during the campaign period.

## 2.6 Data processing and analysis

Deming's total least squares regression was used to compare measurements from different methods or instruments, estimate the mass absorption coefficients (MAC), and evaluate correlations among variables (R package "deming" Therneau, 2018). Deming regression fits a couple of variables considering the independent measurement errors of both X and Y. The errors are assumed to be normally distributed, and the error ratio ($E_{ratio}$) is constant (Deming, 1943). For $E_{ratio} = 1$, the regression results are equivalent to the orthogonal regression. Measurement data and calculations were processed in the software R version 4.0.0 (R Core Team, 2020).

## 3 Results and Discussion

### 3.1 Aerosol accumulation driven by atmospheric stability

Extremely high values for aerosol mass and particle number concentrations and light absorption coefficients were measured in Retje, Loški Potok, under conditions of near-ground temperature inversion. Figure 3 shows the time series of the OA and BC mass concentrations (Fig. 3a and Fig.3b), the total light absorption coefficient at 370 nm (Fig. 3c), and the particle number size distribution (Fig. 3f) from the campaign period. For conditions of strong inversion, the OA and BC concentrations ranged from 1.9 to 270 µg m$^{-3}$ and 0.1 to 24 µg m$^{-3}$, respectively. In contrast, when the atmosphere was unstable, the OA and BC fluctuated from 0.17 to 120 µg m$^{-3}$ and 0.0 to 8.1 µg m$^{-3}$. The mean total light absorption coefficient at 370 nm was 260 Mm$^{-1}$ (max. 1100 Mm$^{-1}$) under strong inversion, while the mean value registered under an unstable atmosphere was 56 Mm$^{-1}$ (max. 330 Mm$^{-1}$). The mean total particle number concentration (10–600 nm) was 17x10$^3$ particles cm$^{-3}$ under strong inversion, compared with the mean concentration of 6.1x10$^3$ particles cm$^{-3}$ for unstable atmosphere (see Table 3).

Table 4 shows the statistical summary for the PM$_{10}$, EC, OC, and levoglucosan mass concentrations. For these, the 12-hour predominant atmospheric stability was also identified for periods from 06:00 to 18:00 and from 18:00 to 06:00 LT. During periods when strong inversions prevailed, the mass concentrations of PM$_{10}$, OC, and EC were the highest, reaching mean values of 110 µg m$^{-3}$ (max. 200 µg m$^{-3}$), 47 µg m$^{-3}$ (max. 94 µg m$^{-3}$), and 6.4 µg m$^{-3}$ (max. 15 µg m$^{-3}$), respectively. The wood-burning tracer levoglucosan also exhibited higher concentrations during predominant temperature inversion periods, reaching a mean concentration of 6.7 µg m$^{-3}$ (max. 17 µg m$^{-3}$).

The 1-hour- and 12-hour measurements showed decreased values during weak inversion, neutral, and unstable atmosphere periods. Tables 3 and 4 show the statistical summaries of the aerosol measurements by categories of atmospheric stability and for the entire campaign period.

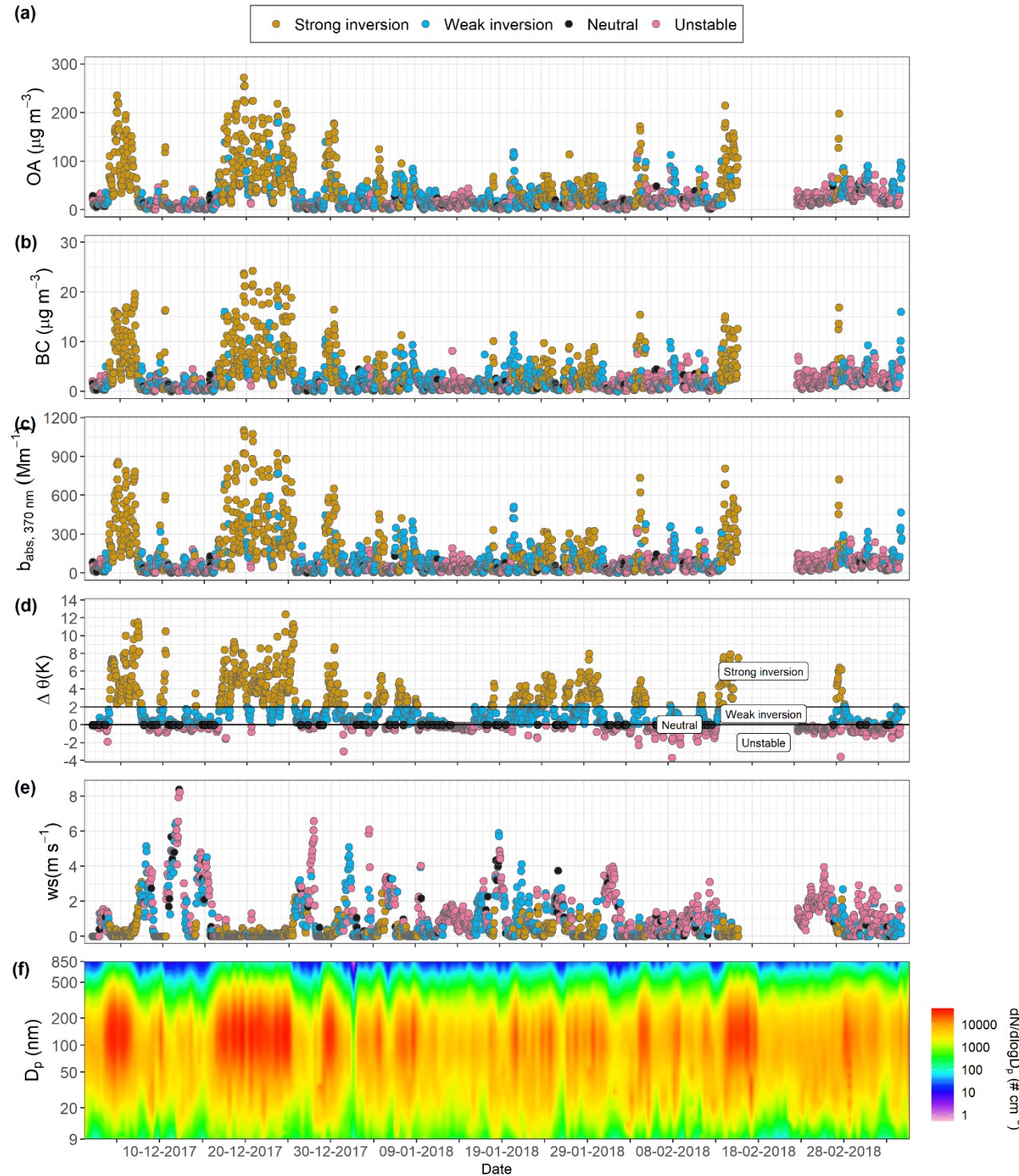

**Figure 3: Time series of 1-hour organic aerosol (a) and black carbon (b) mass concentrations, (c) light absorption coefficient at 370 nm, (d) potential temperature gradient, (e) wind speed, and (f) particle number size distribution at Retje (main station).**

Episodes of strong temperature inversion are of primary interest in this study since they favor the accumulation of aerosol particles produced by a predominant source: RWB. Presumably, the concentrations measured in the village at the bottom of the hollow are less impacted by external aerosol sources under temperature inversion. Therefore, its characterization gives an authentic fingerprint of residential wood burning emissions in a real environment. Strong and weak temperature inversion appeared 28 % and 31 % of the time based on hourly measurements, respectively. The more extended episodes of prevailing thermal inversion occurred from 03 to 07 December 2017, 17 to 26 December 2017, 29 December 2017 to 01 January 2018, and 14 to 17 February 2018.

The effect of atmospheric stability on the accumulation of aerosol particles in the hollow is observed when comparing the village and the background measurements. Figures 4 and 5 show the differences ($\Delta$) in the particle number (N) and aerosol mass concentrations by atmospheric stability. Please note that deltas in the number concentration were calculated using ~600 nm as the upper limit of both size distributions (main and background stations) for comparability. The values of $\Delta N$ were, in general, positive and largest under strong inversion, while the lowest and closer to zero values were observed for unstable atmosphere. From Fig. 4, it is evident the significant contribution of RWB emissions for particles in the size range of 50 to 150 nm, with a median $\Delta N_{50-150} = 47$ x $10^2$ particles cm$^{-3}$, 22 x $10^2$ particles cm$^{-3}$, and 15 x $10^2$ particles cm$^{-3}$ under strong inversion, weak inversion, and unstable atmosphere, respectively. In contrast, the median $\Delta N_{10-50}$ were 12 x $10^2$ particles cm$^{-3}$, 9.8 x $10^2$ particles cm$^{-3}$, and 9.6 x $10^2$ particles cm$^{-3}$, and medians for $\Delta N_{150-600}$ were 24 x $10^2$ particles cm$^{-3}$, 9.4 x $10^2$ particles cm$^{-3}$, and 5.3 x $10^2$ particles cm$^{-3}$, for each category of atmospheric stability, respectively. In general, $\Delta N_{10-50}$ exhibited the lower change among the stability conditions, given that from strong inversion to weak inversion, $\Delta N_{50-100}$ reduced by 50 % and $\Delta N_{100-600}$ by 60 %. We hypothesize that the almost constant $\Delta N_{10-50}$ is explained by the predominant sources of ultrafine particles: secondary aerosol particles, sea salt, and traffic emissions (Leoni et al., 2018). All these three sources might have an impact on the local and background concentrations at the study site. $\Delta BC$ showed similar trends to $\Delta N$. For strong inversion, weak inversion, and unstable atmosphere, the mean $\Delta BC$ was 4.4 µg m$^{-3}$, 1.5 µg m$^{-3}$, and 0.9 µg m$^{-3}$, respectively. In some cases, the values of $\Delta BC$ were negative, meaning the concentrations in the background were higher than in the village.

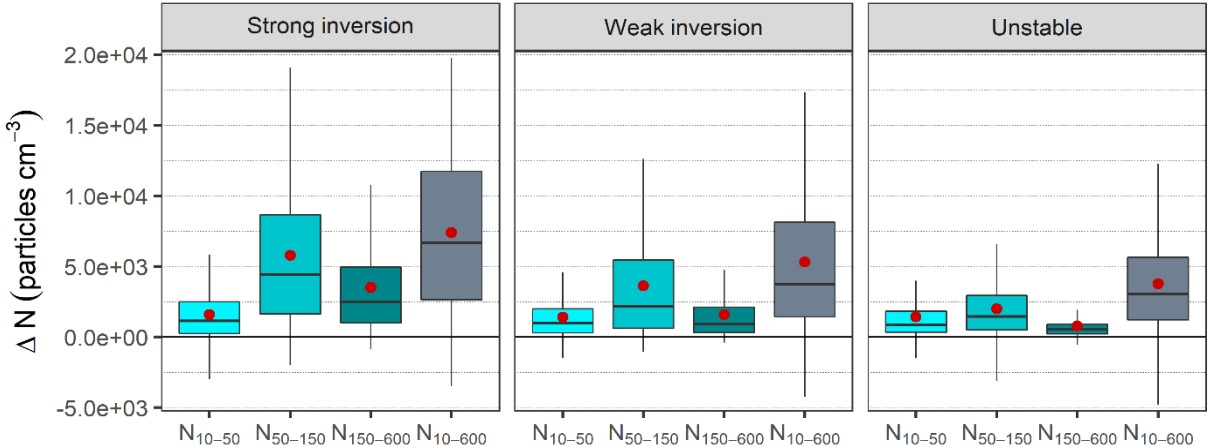

**Figure 4: Boxplots and mean values (red dots) of the differences in N between the village and the background stations, according to size ranges.** The maximum size range in the village was set to 600 nm to match the maximum measurement limit from the MPSS at the background station. The lower and upper borders of the boxes represent the first and third quartiles on which the middle 50 % of the statistical variables are located, the black horizontal lines inside the boxes represent the median, and the whiskers represent the minimum and maximum values without outliers.

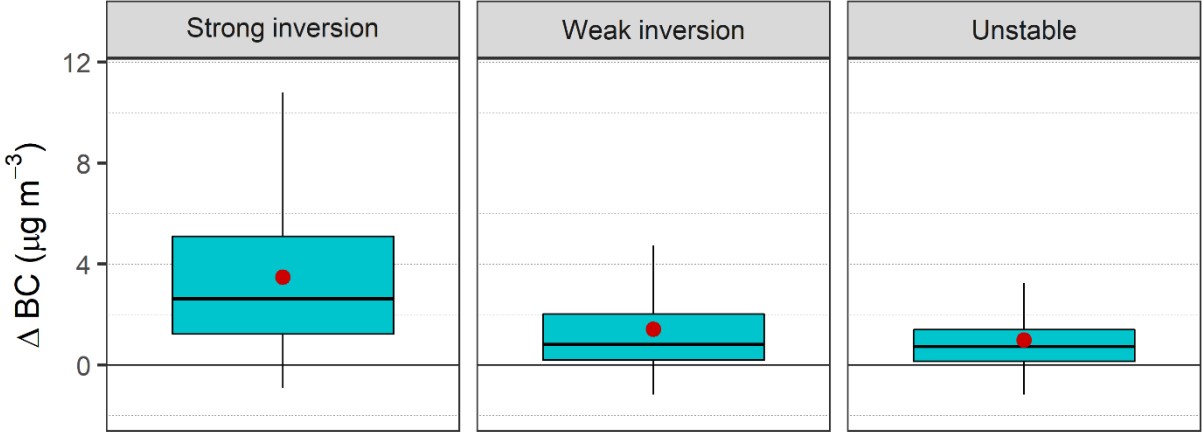

**Figure 5: Boxplots and mean values (red dots) of the differences in BC between the village and the background stations, according to atmospheric stability.** The lower and upper borders of the boxes represent the first and third quartiles on which the middle 50 % of the statistical variables are located, the black horizontal lines inside the boxes represent the median, and the whiskers represent the minimum and maximum values without outliers.

### 3.2 Aerosol optical properties

### 3.2.1 Light absorption coefficients

In the village, the aerosol light absorption coefficient at 370 nm ($b_{abs}$ (370 nm)) reached 1100 $Mm^{-1}$ (under strong inversion). The total light absorption measured by the AE33 at multiple wavelengths ($b_{abs}(\lambda)$) was assumed to include the contribution of

both BC and BrC. To apportion the light absorption corresponding to BC and BrC at each wavelength, we used a method based on the absorption Ångström exponent (Lack and Langridge, 2013; Massabò et al., 2015). Accordingly, for a given wavelength, we have:

$$b_{abs}(\lambda) = b_{abs,BC}(\lambda) + b_{abs,BrC}(\lambda) ,$$ (9)

For a given pair of wavelengths $\lambda_1$ and $\lambda_2$, the mathematical definition of the BC absorption Ångström exponent ($AAE_{BC}$) can be written as:

$$\frac{b_{abs,BC}(\lambda_1)}{b_{abs,BC}(\lambda_2)} = \left(\frac{\lambda_1}{\lambda_2}\right)^{-AAE_{BC}} ,$$ (10)

Taking $\lambda_2$ as 950 nm and assuming that at this wavelength, the total absorption corresponds entirely to BC (the contribution from BrC is negligible) and that $AAE_{BC} = 1$, Eq. 10 is rearranged as follows,

$$b_{abs,BC}(\lambda_1) = b_{abs}(950 \ nm) * \left(\frac{\lambda_1}{950}\right)^{-1} ,$$ (11)

Combining Eq. 9 and Eq. 11, the BrC absorption at $\lambda_1$ will be:

$$b_{abs,BrC}(\lambda_1) = b_{abs}(\lambda_1) - b_{abs,BC}(\lambda_1),$$ (12)

In Eq. 12 $\lambda_1$would be any wavelength between 370 nm and 880 nm.

The contribution of $b_{abs, BrC}$ to the total absorption decreased toward the infrared, where the measured light absorption was assumed to be 100 % from BC. The values of $b_{abs}$ (950 nm) reached 160 Mm$^{-1}$, 110 Mm$^{-1}$, and 54 Mm$^{-1}$ for strong inversion, weak inversion, and unstable atmosphere, respectively.

A significant contribution from BrC to the total aerosol light absorption was measured in Loški Potok: during strong inversion, the average contribution of $b_{abs, BrC}$ (370 nm) was 60 % (range = 17–78 %, Fig. 6). For conditions of weak inversion and unstable atmosphere, the average contribution of $b_{abs, BrC}$ (370 nm) was 54 % and 43 %, respectively. The contribution of $b_{abs, BrC}$ to the total absorption decreased toward the infrared (470 nm = 46 %, 520 nm = 37 %, 590 nm = 29 %, 660 nm = 18 %, and 880 nm = 3 %).

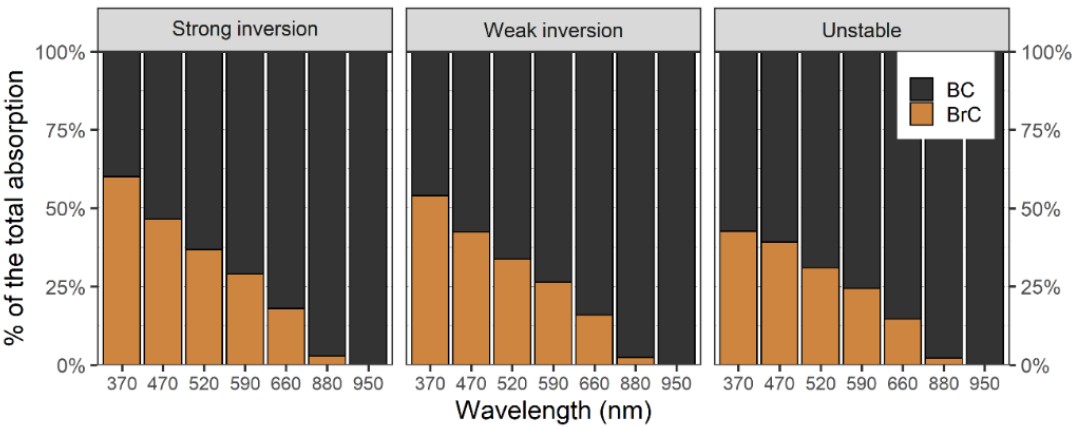

**Figure 6: Mean BC and BrC contributions to light absorption from 370 nm to 950 nm in Retje, Loški Potok, according to atmospheric stability. Results for neutral atmosphere are not shown due to insufficient data.**

The average contribution of BrC to the total aerosol light absorption in the near UV under strong inversion is significantly higher than findings from previous studies in urban and rural locations impacted by wood and biomass burning emissions. For instance, the national study by Zhang et al. (2020) in France found a maximum BrC contribution of 42 % in the Paris area. Mbengue et al. (2021) estimated that BrC contributions in a rural and regional background station in the Czech Republic reach 19 % in winter. In the Indian city of Kanpur, the mean contribution for BrC absorption was 30 % (Shamjad et al., 2016). In a suburban site near Guangzhou in China, BrC contributed 24 % of the total UV absorption (Qin et al., 2018). A similar and substantial contribution from BrC to the total light absorption was found in the city of Ioannina (110.000 inhabitants, (POCITYF, 2023)), Greece, where severe RWB emissions produced an average BrC contribution of 68 % (calculated from the reported mean values of $b_{abs}$ and $b_{abs, BrC}$ at 370 nm, (Kaskaoutis et al., 2022)).

### 3.2.2 BrC (OA) absorption Ångström exponent

The wavelength-dependence of $b_{abs, BrC}$ is plotted in Fig. 7a. Initially, a power law fit was applied to the spectral range 370–880 nm; nevertheless, we observed that the fitting curve for $b_{abs, BrC}$ throughout 370–880 nm resulted in an overestimation of the absorption at 370 nm of about ~50 % (dotted grey line in Fig. 7a). Similar findings have been observed in other studies (Hoffer et al., 2006) and are associated to the presence of internally mixed aerosol particles, whose content of BrC affects the different spectral dependencies (Kumar et al., 2018). Using one single $AAE_{BrC}$ might not be representative, and some studies suggest that $AAE_{BrC}$ is wavelength-dependent (Hoffer et al., 2006; Utry et al., 2014). Utry et al. (2014) found better correlations between particle modes and geometric mean diameters, levoglucosan/total carbon ratio, and OC/EC ratio with $AAE_{BrC}$ computed for the range of 355 to 532 nm; in contrast, comparatively poorer correlations were found when $AAE_{BrC}$ was estimated for the range 266 to 1064 nm. Consequently, we recalculated $AAE_{BrC}$ for two separate ranges: 370–590 nm (more significant OA absorption) and 590–880 nm (lower OA absorption) and selected those AAE for which $R^2 \geq 0.8$. Figure 7a shows the substantial change in the slope of the fitted lines for both ranges of wavelengths. The resulting $AAE_{BrC}$ were $AAE_{BrC}$

$_{370–590\ nm}$ = 3.9 (±SD 0.4) and AAE$_{BrC\ 590–880\ nm}$ = 7.6 (±SD 0.4) under strong inversion (Fig. 7b and 7c). Multiple authors have addressed the sensitivity of the AAE to the range of wavelengths selected for its calculation (Harrison et al., 2013; Utry et al., 2014). Yet, the extent of this sensitivity depends on the aerosol source. It is lower for aerosols with predominant BC (Cuesta-Mosquera et al., 2021) and higher for samples containing a substantial contribution of organic species.

The median values of AAE differ by atmospheric stability (statistically significant difference, non-parametric median Mood's

test, *p-values* < 0.05, α = 0.05). Under weak inversion and unstable atmosphere, AAE$_{BrC\ 370–590\ nm}$ was 3.7 (±SD 0.5) and 3.5 (±SD 0.5), and AAE$_{BrC\ 590–880\ nm}$ was 7.8 (±SD 0.7) and 7.7 (±SD 1.1), respectively. The variation between the median AAE values amid the atmospheric stability categories suggests a slight change in the composition of the organic aerosols among the three, most likely driven by the mixing of aerosols from external sources and photochemical processes (He et al., 2022; Liu et al., 2016).


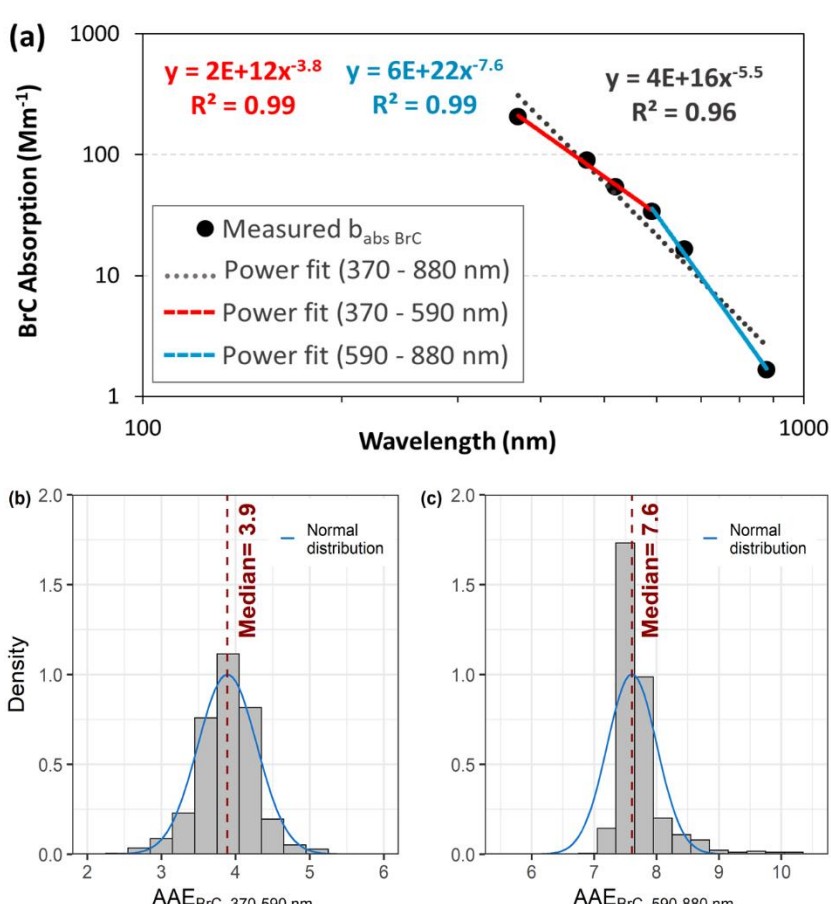

**Figure 7: (a) Power law fittings of the BrC absorption spectra in log-log scale, (b) histogram of the BrC Absorption Ångström Exponent from 370 to 590 nm, and (c) histogram of the BrC Absorption Ångström Exponent from 590 to 880 nm.**

### 3.2.3 BrC (OA) and BC Mass absorption cross-sections (MAC)

The RWB OA mass absorption cross-section ($MAC_{OA}$) was estimated as the slope from the fitting among the organic aerosol mass and the apportioned BrC light absorption. Figure 8a shows the scatter plot and orthogonal fitting for measurements under strong inversion (highest accumulation of OA).

For the study site, the estimated RWB $MAC_{OA, 370\,nm}$ = 2.4 $m^2$ $g^{-1}$ (calculated for conditions of strong inversion). For periods of weak inversion $MAC_{OA, 370\,nm}$ did almost not change (2.3 $m^2$ $g^{-1}$). On the contrary, for an unstable atmosphere, $MAC_{OA, 370\,nm}$ 

$_{nm}$ (1.8 $m^2$ $g^{-1}$), was lower and statistically significantly different than the MAC values calculated for strong and weak inversion (hypothesis test, $p$-value < 0.0001, $\alpha$ = 0.05). The decreased $MAC_{OA, 370\,nm}$ for unstable conditions, is associated with lower OA absorption. Along with little absorption, the reduced $MAC_{OA, 370\,nm}$, indicates the presence of diverse light-absorbing compounds during an unstable atmosphere since the mixing with aerosols from external non-local sources and regionally processed aerosol is presumed.

Table 5 shows the spectral variation of $MAC_{OA}$ calculated by correlating the OA mass and the BrC apportioned absorption at multiple wavelengths (370 nm, 470 nm, 520 nm, 590 nm, and 660 nm). The major $MAC_{OA}$ value was calculated for 370 nm. We observed reduced $MAC_{OA}$ by increasing wavelength due to decreased OA light absorption when moving toward the infrared. It was also observed that for larger wavelengths, the variability of the $MAC_{OA}$ within the atmospheric stability categories was insignificant as $b_{abs, BrC}$ approached zero.

The same approach was used to estimate $MAC_{BC}$ by correlating the EC mass from filters and the 12-hour averaged absorption at 950 nm. For the study site, $MAC_{BC, 950\,nm}$ = 6.7 $m^2$ $g^{-1}$ (Fig. 8b). For periods of weak inversion and unstable conditions, $MAC_{BC, 950\,nm}$ was 6.5 $m^2$ $g^{-1}$ and 7.2 $m^2$ $g^{-1}$, respectively. Although the calculated $MAC_{BC, 950\,nm}$ for unstable conditions seems to be slightly higher, we found that the differences among the slopes are not statistically significant, i.e., there is no interaction effect from the atmospheric stability (strong inversion and unstable atmosphere: hypothesis test, $p$-value = 0.114, $\alpha$ = 0.05;

weak inversion and unstable atmosphere: hypothesis test, $p$-value = 0.088, $\alpha$ = 0.05).

We compared our specific RWB $MAC_{OA}$ to values reported in the literature for wood and biomass-burning emissions. In the near-UV region, the values of $MAC_{OA}$ are spread and range between 0.2 and 5.8 $m^2$ $g^{-1}$ (Table 6). Comparable results to those from Loški Potok were observed in the studies of Chen and Bond (2010) for laboratory analyses of methanol-soluble organic carbon (MSOC) from wood combustion ($MAC_{MSOC, 370\,nm}$ = 2.0 $m^2$ $g^{-1}$) and Cheng et al. (2011) for water-soluble organic

carbon extracts (WSOC) from an urban environment in Beijing impacted by biomass burning and fossil fuel emissions during winter ($MAC_{WSOC, 370\,nm}$ = 1.8 $m^2$ $g^{-1}$). The spreading of the reported $MAC_{OA}$ values originates in the OA nature; these are heterogeneous and non-separable mixtures of absorbing and non-absorbing materials, whose composition and physicochemical properties change depending on (i) how the OA is constrained and measured, (ii) the burning source, (iii) the combusting conditions, (iv) the geographical location and time of measurements, and (v) the meteorological conditions. Points (iv) and (v)

influence the OA aging driven by photochemical reactions in the atmosphere and the presence of aerosols from distinct combustion sources. An example to illustrate (i) refers to the unequal light-absorbing properties of the WSOC and the MSOC

extracted from particulate matter filters (Chen and Bond, 2010; Kim et al., 2016; Zhu et al., 2018). In addition, whether the OA should be analyzed as a fraction or a bulk has been addressed previously and remains an open discussion (Zhang et al., 2016). These facts reinforce the necessity of performing specific and source-oriented OA characterization.


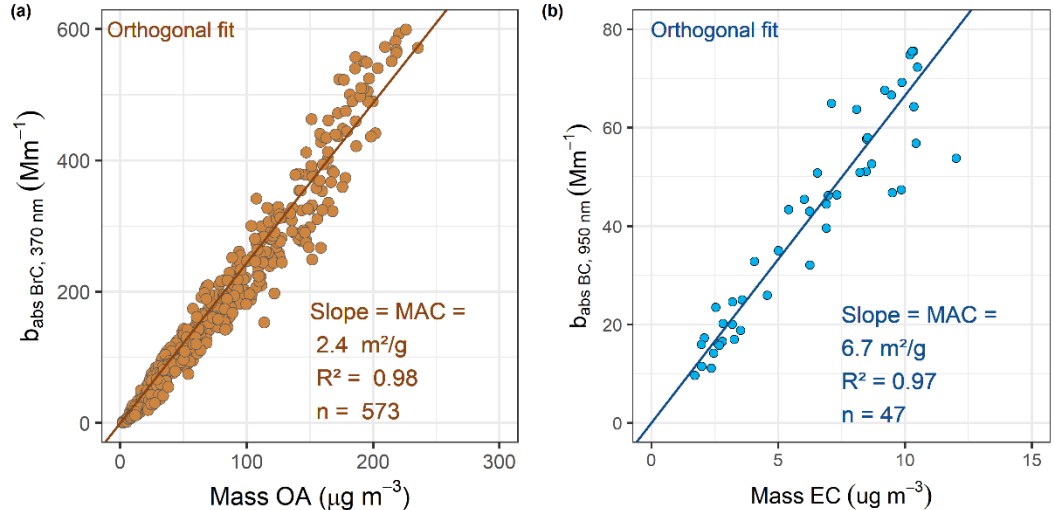

**Figure 8: Scatter plots and orthogonal fits for the village measurements of (a) 1-hour BrC light absorption at 370 nm and OA mass concentrations and (b) 1-hour BC light absorption at 950 nm and EC mass concentrations under strong near-ground temperature inversion.** The figure includes the coefficient of determination ($R^2$), and the number of observations (n). The intercept was forced through
zero.

**Table 5. Spectral variation of the organic aerosol MAC by atmospheric stability.**

| Wavelength | $MAC_{OA}$ ($m^2$ $g^{-1}$) | | |
|---|---|---|---|
| | Strong inversion | Weak Inversion | Unstable |
| 370 nm | 2.4 | 2.3 | 1.8 |
| 470 nm | 1.0 | 1.0 | 0.8 |
| 520 nm | 0.4 | 0.4 | 0.3 |
| 590 nm | 0.3 | 0.3 | 0.2 |
| 660 nm | 0.2 | 0.2 | 0.1 |
| 880 nm | < 0.1 | < 0.1 | < 0.1 |
| 950 nm | Zero by definition | | |

**Table 6. Comparison of light absorption properties for OA (also referred to as BrC and OC) reported in the literature.**

| Type of study | Reference and region | MAC ($m^2$ $g^{-1}$) | | AAE | | Type of site and main aerosol sources |
|---|---|---|---|---|---|---|
| | | $\lambda$ (nm) | Value | $\lambda$ (nm) | Value | |

| Laboratory | Chen and Bond, 2010 | 370 | 0.6[a] | 360-500 | 9.4 | Wood (pine) |
| | | | 2.0[b] | | 7.5 | |
| | Olson et al., 2015 | 370 | 0.21 | n.a. | n.a. | Wood pellets |
| | Kumar et al., 2018 | 370 | 5.5 | 370, 880 | 4.6 | Beechwood |
| Field campaign | Cheng et al., 2011 (West Asia) | 365 | 1.8 | 330-480 | 7.5 | Urban; BB, FF |
| | Zhang et al., 2021 (West Asia) | 370 | 4.3 | 370-520 | 4.2 | Urban; CC, FF, RWB |
| | Srinivas and Sarin, 2014 (South Asia) | 365 | 0.78 | 300-800 | 6.0 | Urban; BB, RWB, FF |
| | Kim et al., 2016 (East Asia) | 365 | 0.85[c] | 300-700 | 5.1[d] | Urban background; BB, FF, LRT |
| | | | 1.02[a] | | 7.2[e] | |
| | Chen et al., 2020 (East Asia) | 365 | 0.91[a] | n.d. | n.d. | Urban; BB, FF, LRT |
| | | | 1.1[b] | | | |
| | Zhang et al., 2016 (North America) | 405 | 0.60 | n.d. | n.d. | Urban; BB, FF, RWB |
| | Lack et al., 2012 (North America) | 404 | 1.0 | 404-658 | 2.3 | Rural; FoF |
| | Hoffer et al., 2006 (South America) | n.a. | n.a. | 300-700 | 6.4 | Rural; FoF |
| | Liakakou et al., 2020 (South Europe) | 370 | 4.3 | 370-660 | 3.9[f] | Urban background; FF, RWB |
| | **This study, 2023 (Central Europe)** | **370** | **2.4** | **370-590** | **3.9** | **Rural; RWB** |
| | | | | **590-880** | **7.6** | |
| | | | | **370-880** | **5.5** | |

a: WSOC; b: MSOC; c: OC; d. Wavelength-dependency of the absorption measured on MSOC; e. Wavelength-dependency of the absorption measured on WSOC; f: Wintertime; BB: Biomass burning other than wood burning; CC: Coal combustion; FF: Fossil fuel; FoF: Forest fires; LRT: Long-range transport; RWB: Residential wood burning; n.a.: non-applicable data; n.d.: non-available data.

### 3.3 Analysis of uncertainty

Error propagation was used to estimate the uncertainty of the RWB MAC$_{OA}$, considering the contribution of the OA mass concentrations (OA) and the BrC light absorption coefficients. The mathematical expression representing the fit between both

variables has the form y = m*x, where y = $b_{abs, BrC}$ ($\lambda$), x = OA, and m = $MAC_{OA}$. Rearranging the equation, $MAC_{OA} = f = y/x$. For uncorrelated variables, the propagated error of the function ($\sigma_f$) can be calculated using the following quadratic sum:

$$\sigma_f = \sqrt{\left(\frac{\delta f}{\delta x}\sigma_x\right)^2 + \left(\frac{\delta f}{\delta y}\sigma_y\right)^2}, \tag{13}$$

Where $\sigma_x$ and $\sigma_y$ are the absolute errors of x and y.

To estimate the $MAC_{OA}$ uncertainty, the propagated errors of $b_{abs, BrC}$ ($\lambda$) and OA were first calculated individually. Based on the OA mass balance (Eq. 2), the uncertainty of the OA mass concentration includes the contribution of the following individual components: (i) $PM_1$ mass concentrations estimated from the MPSS, (ii) the eBC mass concentrations derived from the AE33, and (iii) the mass concentration of inorganic species in the $PM_{10}$ filters. The $PM_1$ uncertainty was assumed to be 17 % based
on the calculations of Buonanno et al. (2009) and the MPSS intercomparison in the laboratory. This estimation accounts for contributions from the sampling flow rate, the volumetric diameter, the diffusion efficiency corrections, and the particle density. The uncertainty of the eBC mass concentration is assumed to be 5 %, corresponding to the EC uncertainty, as eBC was normalized to EC using the local $MAC_{BC}$. The uncertainty of EC corresponds to the reproducibility of the thermo-optical analysis following the EUSAAR protocol. The uncertainty contribution of the insoluble inorganics was 20 %, considering the
deviation resulting from assuming the InA fraction in $PM_{10}$ to be the same as that in $PM_1$. Additionally, we considered a reproducibility error based on Leiva et al. (2012).

In the uncertainty calculation for $b_{abs, BrC}$ ($\lambda$) the following contributions were considered: (i) the AE33 light absorption coefficients, (ii) the harmonization factor $H$, and (iii) the BC absorption Ångström exponent. The uncertainty of the AE33 total light absorption coefficients is assumed to be 25 % and mainly proceeds from the variabilities in the unit-to-unit
intercomparison and the multiple light scattering correction factor $C$ (Cuesta-Mosquera et al., 2021; Müller et al., 2011; WMO, 2016). Given the presence of correlated variables in the mathematical definition of $b_{abs, BrC}$ ($\lambda$), the uncertainty calculation included the covariance when dealing with total light absorption coefficients at different wavelengths measured by the AE33. The propagated uncertainty of $H$ was calculated based on the simultaneous AE33-MAAP field measurements in Melpitz, Germany, during winter time (2018-2019). For this, the uncertainty of the MAAP light absorption coefficients of 12 % was
included (Petzold and Schönlinner, 2004). The calculated uncertainty of $H$-winter for Melpitz (median = 1.9) is 23 % and assumed constant for the study in Loški Potok. The uncertainty of the BC absorption Ångström exponent was assumed as 10 %, coming from the deviation of the $AAE_{BC}$ values ranging between 0.8 and 1.2 (Lack and Langridge, 2013). Although the assumption of AAEBC = 1 might add significant error to the apportionment of the light absorption coefficients (BC/BrC) and aerosol optical properties, the value was used as a fair estimate based on the community standards. The values of uncertainty
estimated for $b_{abs, BrC}$ ($\lambda$) were 48 % (370 nm), 26 % (470 nm), 19 % (520 nm), 14 % (590 nm), 11 % (660 nm), and 6 % (880 nm).

The final propagated uncertainty for the $MAC_{OA, 370 \, nm}$ was 46 %, with a major dependency on the apportioned OA light absorption coefficient. The uncertainty of $MAC_{OA}$ increased with the wavelength due to lower $MAC_{OA}$ values and light absorption coefficients attributed to BrC; for instance, the uncertainty of $MAC_{OA, 470 \, nm}$ was 60 % and > 70 % for $MAC_{OA, 520 \, nm}$. Similarly, the uncertainty of the $PM_1/PM_{10}$ ratio (slope in Fig. 2) was calculated through error propagation. Besides the contribution of $PM_1$, the error for $PM_{10}$ was considered and assumed to be 10 %, accounting for sampling errors (air volume) and reproducibility (Hafkenscheid, 2013; Leiva et al., 2012). The final determined uncertainty for the slope $PM_1/PM_{10}$ was 15 %.

### 3.4 Climate impact: sensitivity analysis through simple forcing efficiency calculations

The model described in section 2.4 was used as a sensitivity analysis tool to assess the climate impact of the light-absorbing properties of the RWB aerosol particles at Loški Potok. The SFE represents the perturbation to the atmospheric radiative balance of the Earth by a given mass of aerosols (Choudhary et al., 2021). Figure 9 shows the spectral variation of the SFE for two different types of surfaces: fresh snow ($a_s$ = 0.80) and Earth-average ($a_s$ = 0.19). In each case, the forcing efficiency was calculated for two scenarios: (i) assuming OA as light-absorbing species (imaginary refractive index $k_{BrC}$ from the literature) and (ii) assuming OA as non-absorbing (imaginary refractive index $k_{BrC}$ = 0). The SFE was calculated for six of the seven wavelengths covered by the AE33. The total radiative forcing (RF) was calculated by integrating the area under the curves using the trapezoidal rule (blue and yellow lines in Fig. 9).

The resulting SFE was larger and positive for bright snow and relatively small for the Earth-average surface. For non-absorbing OA, the SFE was lower than for absorbing with slightly negative values for the Earth-average surface. The comparison among surfaces indicates that RWB aerosols have a more significant impact over bright surfaces, for instance, the Arctic or areas covered by snow, typical during the coldest season when the most significant emissions of OA are expected due to more intense RWB. We also observed that SFE is larger at 470 nm despite the significantly higher mass absorption cross-section calculated at 370 nm. This observation is attributed to the larger solar irradiance $dS(\lambda)/d(\lambda)$ reported in the blue region (1.1 to 1.4 W m$^{-2}$ nm$^{-1}$) compared to the values from the near UV (0.0 to 0.8 W m$^{-2}$ nm$^{-1}$).

For fresh snow, the corresponding RF of the RWB aerosols was 61 W g$^{-1}$ and 44 W g$^{-1}$ for absorbing OA and non-absorbing OA, respectively. For an Earth-average surface, the RF was 0.2 W g$^{-1}$ and -3.9 W g$^{-1}$ for absorbing and non-absorbing OA. The RF was two times higher for a snow-like area and passed from cooling to warming for an Earth-average surface by taking OA as light-absorbing species. Our calculations show the crucial influence of the OA in atmospheric warming via the lensing effect of coated BC and BrC absorption in shorter wavelengths; notwithstanding, robust climate models are essential to reporting definite radiative forcing figures since these consider further atmospheric processes and aerosol properties such as aerosol-cloud interactions, aerosol hygroscopicity, and the total vertical column of aerosols.

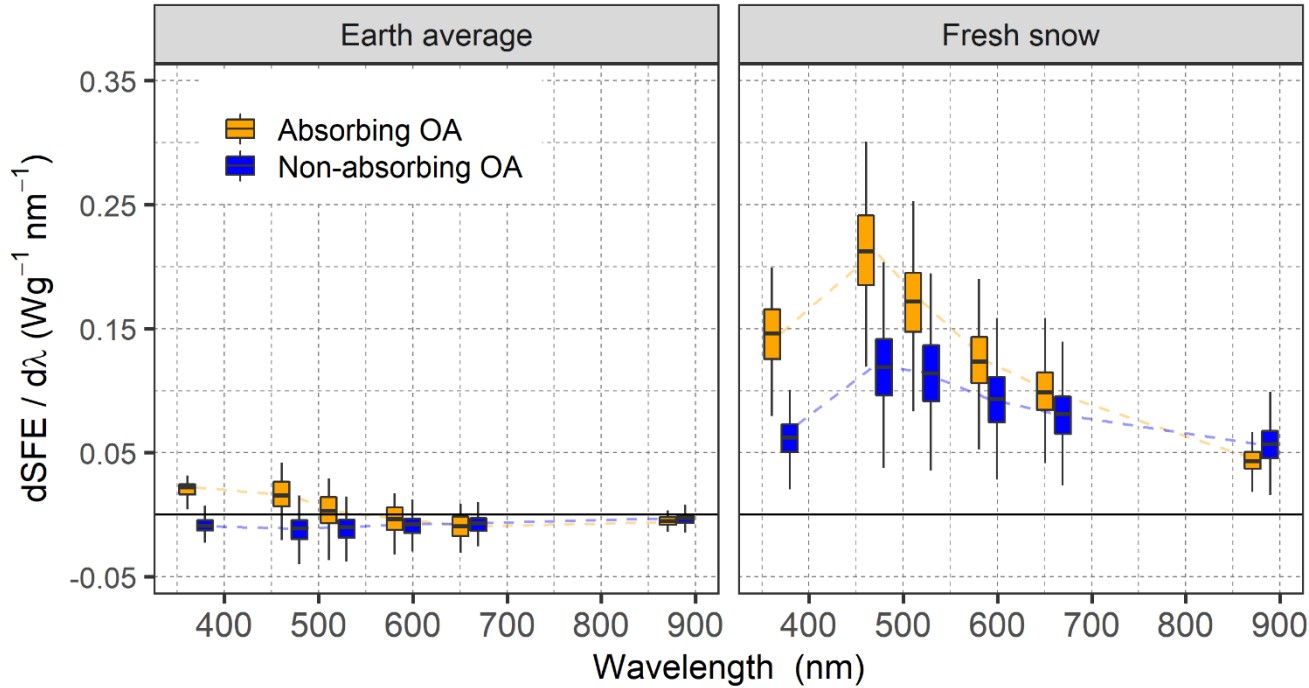

**Figure 9: Spectral variation of the simple forcing efficiency of the RWB aerosols characterized at Loški Potok and comparison of results obtained from considering absorbing OA and non-absorbing OA for two types of surfaces: fresh snow and Earth-average.** The yellow and blue dotted lines connect the median values of dSFE/dλ.

**Table 3. Statistics for 1-hour measurements of mass concentrations, particle number concentration, and light absorption coefficients at the village Retje, according to atmospheric stability.** The statistics in parentheses correspond to the 10th (P10) and 90th (P90) percentiles.

| Atmospheric stability | Strong inversion | | | Weak inversion | | | Neutral | | |
|---|---|---|---|---|---|---|---|---|---|
| % Occurrence during the campaign | 28 % | | | 31 % | | | 5 % | | |
| Measurement | Min (P10) | Max (P90) | Mean ± SD | Min (P10) | Max (P90) | Mean ± SD | Min (P10) | Max (P90) | Mean ± SD |
| $PM_1$ (µg m⁻³) | 2.4 (17) | 350 (200) | 91 ± 71 | 0.4 (4.3) | 230 (73) | 32 ± 33 | 0.8 (2.4) | 68 (38) | 17 ± 16 |
| OA (µg m⁻³) | 1.9 (13) | 270 (160) | 71 ± 56 | 0.2 (3.4) | 180 (56) | 25 ± 26 | 0.6 (1.7) | 55 (30) | 14 ± 13 |
| BC (µg m⁻³) | 0.1 (0.87) | 24 (14) | 6.1 ± 5.0 | 0.0 (0.19) | 17 (5.2) | 2.3 ± 2.4 | 0.0 (0.08) | 4.7 (3.2) | 1.2 ± 1.1 |
| TC (µg m⁻³) | 3.6 (8.5) | 130 (67) | 34 ± 27 | 3.2 (4.7) | 110 (38) | 18 ± 16 | 3.5 (3.8) | 21 (17) | 9.8 ± 5.0 |

| Measurement | Min (P10) | Max (P90) | Mean ± SD | Min (P10) | Max (P90) | Mean ± SD | Min (P10) | Max (P90) | Mean ± SD |
|---|---|---|---|---|---|---|---|---|---|
| Total N (x$10^3$ cm$^{-3}$; 100-600 nm) | 1.3 (4.8) | 54 (35) | 17 ± 12 | 0.54 (1.8) | 59 (18) | 8.5 ± 7.5 | 0.41 (1.2) | 17 (9.4) | 4.9 ± 3.5 |
| $b_{abs\ 370\ nm}$ (Mm$^{-1}$) | 2.1 (32) | 1100 (590) | 260 ± 230 | 0.2 (6.2) | 770 (200) | 89 ± 99 | 0.1 (3.0) | 160 (100) | 43 ± 41 |
| $b_{abs\ BrC\ 370\ nm}$ (Mm$^{-1}$) | 0.5 (18.4) | 700 (380) | 160 ± 140 | 0.0 (2.9) | 480 (120) | 51 ± 61 | 0.0 (1.2) | 91 (59) | 22 ± 23 |
| $b_{abs\ BC\ 370\ nm}$ (Mm$^{-1}$) | 1.5 (13) | 410 (230) | 99 ± 85 | 0.2 (3.2) | 290 (88) | 38 ± 41 | 0.4 (1.3) | 80 (52) | 21 ± 20 |
| $b_{abs\ 880\ nm}$ (Mm$^{-1}$) | 0.7 (5.6) | 180 (99) | 43 ± 37 | 0.10 (1.4) | 130 (38) | 16 ± 18 | 0.20 (0.57) | 34 (22) | 9.1 ± 8.8 |
| $b_{abs\ BrC\ 880\ nm}$ (Mm$^{-1}$) | 0.0 (0.14) | 6.4 (3.2) | 1.3 ± 1.3 | 0.0 (0.0) | 4.5 (0.94) | 0.4 ± 0.5 | 0.0 (0.0) | 0.9 (0.47) | 0.2 ± 0.2 |
| $b_{abs\ BC\ 880\ nm}$ (Mm$^{-1}$) | 0.70 (5.6) | 170 (96) | 42 ± 36 | 0.10 (1.3) | 120 (37) | 16 ± 17 | 0.20 (0.6) | 34 (22) | 8.9 ± 8.6 |

**Table 3. (Continued) Statistics for 1-hour measurements of mass concentrations, particle number concentration, and light absorption coefficients at the village Retje, according to atmospheric stability.** The statistics in parentheses correspond to the 10th (P10) and 90th (P90) percentiles.

| Atmospheric stability | Unstable | | | Total period | | |
|---|---|---|---|---|---|---|
| % Occurrence during the campaign | 36 % | | | | | |
| Measurement | Min (P10) | Max (P90) | Mean ± SD | Min (P10) | Max (P90) | Mean ± SD |
| PM$_1$ (µg m$^{-3}$) | 0.20 (5.5) | 200 (44) | 23 ± 18 | 0.20 (5.9) | 3508 (110) | 45 ± 52 |
| OA (µg m$^{-3}$) | 0.17 (4.4) | 120 (34) | 18 ± 13 | 0.17 (4.4) | 270 (85) | 35 ± 41 |
| BC (µg m$^{-3}$) | 0.0 (0.24) | 8.1 (3.4) | 1.6 ± 1.3 | 0.0 (0.26) | 24 (7.5) | 3.1 ± 3.6 |
| TC (µg m$^{-3}$) | 3.1 (4.5) | 60 (21) | 12 ± 7.4 | 3.1 (4.9) | 130 (42) | 19 ± 19 |
| Total N (x$10^3$ cm$^{-3}$; 100-600 nm) | 0.44 (1.7) | 49 (11) | 6.1 ± 4.8 | 0.41 (2.0) | 59 (23) | 9.9 ± 9.3 |
| $b_{abs\ 370\ nm}$ (Mm$^{-1}$) | 0.10 (8.4) | 330 (110) | 56 ± 44 | 0.10 (8.9) | 1100 (310) | 120 ± 160 |
| $b_{abs\ BrC\ 370\ nm}$ (Mm$^{-1}$) | 0.0 (3.7) | 190 (58) | 28 ± 25 | 0.0 (4.1) | 700 (190) | 72 ± 100 |
| $b_{abs\ BC\ 370\ nm}$ (Mm$^{-1}$) | 0.10 (4.0) | 140 (57) | 28 ± 22 | 0.10 (4.4) | 410 (120) | 51 ± 60 |

| Measurement | | | | | | |
|---|---|---|---|---|---|---|
| $b_{abs\ 880\ nm}$ (Mm$^{-1}$) | 0.0 (1.7) | 59 (24) | 12 ± 9.5 | 0.0 (1.9) | 180 (52) | 22 ± 26 |
| $b_{abs\ BrC\ 880\ nm}$ (Mm$^{-1}$) | 0.0 (0.0) | 1.6 (0.55) | 0.25 ± 0.24 | 0.0 (0.0) | 6.4 (1.5) | 0.58 ± 0.86 |
| $b_{abs\ BC\ 880\ nm}$ (Mm$^{-1}$) | 0.0 (1.7) | 58 (24) | 12 ± 9.3 | 0.0 (1.8) | 170 (51) | 21 ± 25 |

**Table 4. Statistics for 12-hour measurements of PM$_{10}$ and its composition at the village Retje, according to atmospheric stability.** The numbers in parentheses correspond to the 10th (P10) and 90th (P90) percentiles.

| Atmospheric stability | Strong inversion | | | Weak Inversion | | | Unstable | | | Total period | | |
|---|---|---|---|---|---|---|---|---|---|---|---|---|
| % Occurrence during the campaign | 36 % | | | 33 % | | | 31 % | | | | | |
| Measurement | Min | Max | Mean ± SD | Min | Max | Mean ± SD | Min | Max | Mean ± SD | Min | Max | Mean ± SD |
| PM$_{10}$ (µg m$^{-3}$) | 28 (43) | 200 (170) | 110 ± 51 | 15 (22) | 84 (63) | 37 ± 17 | 13 (20) | 82 (51) | 37 ± 15 | 13 (22) | 200 (150) | 63 ± 49 |
| EC (µg m$^{-3}$) | 1.7 (2.4) | 15 (10) | 6.4 ± 3.3 | 0.37 (0.84) | 5.3 (3.6) | 2.2 ± 1.2 | 0.36 (0.62) | 6.0 (3.5) | 1.8 ± 1.2 | 0.37 (0.85) | 15 (8.5) | 3.6 ± 3.0 |
| OC (µg m$^{-3}$) | 8.6 (14) | 94 (76) | 47 ± 24 | 3.5 (5.7) | 36 (24) | 13 ± 7.9 | 2.2 (4.4) | 24 (15) | 9.6 ± 4.8 | 2.2 (5.9) | 94 (63) | 24 ± 23 |
| Levoglucosan (µg m$^{-3}$) | 1.1 (1.8) | 17 (11) | 6.7 ± 3.9 | 0.33 (0.73) | 7.5 (3.7) | 2.0 ± 1.4 | 0.33 (0.62) | 2.9 (2.1) | 1.3 ± 1.2 | 0.33 (0.78) | 17 (8.9) | 3.5 ± 3.5 |
| K$^{+}$ (µg m$^{-3}$) | 0.44 (0.53) | 5.0 (2.7) | 1.7 ± 0.98 | 0.09 (0.26) | 2.2 (1.1) | 0.56 ± 0.39 | 0.09 (0.19) | 1.2 (0.98) | 0.56 ± 0.28 | 0.09 (0.31) | 5.0 (2.3) | 0.99 ± 0.88 |
| Cl$^{-}$ (µg m$^{-3}$) | 0.07 (0.15) | 1.5 (0.82) | 0.42 ± 0.31 | 0.02 (0.04) | 0.61 (0.35) | 0.17 ± 0.13 | 0.02 (0.04) | 0.67 (0.24) | 0.13 ± 0.14 | 0.02 (0.05) | 1.5 (0.60) | 0.25 ± 0.25 |

## 4 Summary and Conclusions

Residential wood burning (RWB) is currently a significant source of OA and BC emissions; however, the net global cooling and warming effects of aerosols produced by residential biofuel burning it is still not well constrained. To undertake this challenge, further research is needed. In this context, the main focus of this study was the characterization of the aerosol particles produced by intensive RWB in a rural location in Central Europe occupied by 243 households. We evaluated the influence of atmospheric stability on aerosol accumulation, measured and calculated their optical properties using filter-based absorption photometers (Aethalometers AE33), and connected these to climate impact via simple radiative forcing estimates. Intense burning in Loški Potok, Slovenia, and near-ground temperature inversion led to a significant accumulation of aerosols produced by RWB during the coldest season at this rural site. During strong inversion, the mean OA and BC mass concentrations were 71 ± 56 µg m$^{-3}$ and 6.1 ± 5.0 µg m$^{-3}$, respectively, with maximum values of 270 µg m$^{-3}$ and 24 µg m$^{-3}$.

The mean and maximum particle number concentrations were $17 \times 10^3 \pm 12 \times 10^3$ particles $cm^{-3}$ and $54 \times 10^3$ particles $cm^{-3}$, respectively. Deplorable air quality conditions at the bottom of the valley, driven by a stable atmosphere, occurred 60 % of the time in winter. The transition from a stable to an unstable atmosphere reduced OA, BC, and particle number concentrations: from strong inversion to weak inversion and unstable atmosphere, the OA mass concentration decreased by 25 % and 80 %, respectively, the BC mass decreased by 62 % and 74 %, and the particle number concentration decreased by 50 % and 65 %. Significant aerosol accumulation under near-ground temperature inversion and a presumed low mixing were used conveniently to estimate optical properties of RWB under real conditions with low influence from distinct aerosol sources. During strong inversion, the mean $b_{abs\ BrC}$ and $b_{abs\ BC}$ at 370 nm were $160 \pm 140$ $Mm^{-1}$ and $99 \pm 85$ $Mm^{-1}$, respectively. Under weak inversion, $b_{abs,\ BrC}$, and $b_{abs,\ BC}$ were 69 % and 62 % lower, and during unstable atmosphere, 83 % and 73 % lower, respectively. The average contribution of BrC to the total light absorption was 60 % at 370 nm during strong temperature inversion and decreased toward the infrared (470 nm = 46 %, 880 nm = 3 %).

The estimated $MAC_{OA}$ for residential wood burning was 2.4 $m^2\ g^{-1}$ at 370 nm and decreased towards the infrared with values of 1.0 $m^2\ g^{-1}$, 0.4 $m^2\ g^{-1}$, 0.3 $m^2\ g^{-1}$, and 0.2 $m^2\ g^{-1}$, for 470 nm, 520 nm, 590 nm, and 660 nm, respectively. The calculated $MAC_{OA}$ at 880 nm was below 0.1 $m^2\ g^{-1}$. The values of $MAC_{OA}$ for Loški Potok are higher than multiple $MAC_{OA}$ reported from urban and rural locations impacted by wood burning. Nevertheless, differences in MAC between locations are expected due to the influence of diverse aerosol sources and aerosol aging. The higher $MAC_{OA}$ in the study site is attributed to the significant load of organic aerosols and elevated absorption coefficients measured during the campaign period. Nevertheless, the assumptions involved in the calculation of the hourly OA mass concentration are identified as the main limitation and source of uncertainty in the calculation of $MAC_{OA}$. Despite the good agreement found between OA ($OA_{MPSS}$) and $OA_{TCA}$, we recognize that calculating OA from the $PM_1$ mass estimated from the MPSS might lead to an omission of OA fraction in the $PM_1$-$PM_{2.5}$ size range. The RWB $AAE_{BrC}$ was 3.9 in the wavelength range of 370–590 nm. The estimated values of $MAC_{OA}$ and $AAE_{BrC}$ fall within the range of previously reported optical BrC optical properties (Chen and Bond, 2010; Cheng et al., 2011; Liakakou et al., 2020).

The optical properties from the study site were used to estimate SFE and RF from RWB aerosols. The large contribution of BrC to the total aerosol light absorption resulted in a substantial impact. Results showed a turn from cooling to warming for an Earth-average surface when the OA light-absorption properties were considered in the SFE modeling (i.e., the OA were not assumed to be purely scattering compounds). For a snow-covered surface, the warming RF passed from 44 $W\ g^{-1}$ to 61 for absorbing and non-absorbing OA, respectively. This last observation is important since snowy surfaces are common during the coldest seasons and will probably occur in parallel with intense RWB and OA emissions. The estimated SFE and RF were significantly higher than those reported in previous studies such as Chen & Bond (23 $W\ g^{-1}$ over snow and -12 $W\ g^{-1}$ for an Earth average albedo, (2010)). The much larger SFE and RF calculated for Loški Potok are directly related to the substantial $b_{abs}$ measured at the site and the relatively larger MAC, which exceeds typical values reported for other locations impacted by RWB. The SFE and RF calculations were calculated as sensitivity analysis of the aerosol light absorption effect on climate and should be taken as indicative only due to the limitations of this approach, which include the omission of aerosol

hygroscopicity and further atmospheric processes that could take place. Further studies should include more complex atmospheric processes and aerosol properties to report the absolute values for a complete understanding of the climate implications.

As a household-based pollution source, implementing specific technical recommendations may help to prevent and mitigate air pollution produced by residential wood burning. These recommendations include: (i) using updated and certified wood stoves; (ii) managing fuel appropriately by storing the firewood in dry and well-ventilated areas, burn only dry and clean wood, refraining from burning green or treated wood, plastic, trash, clothes or petroleum-derived products; (iii) optimizing burning conditions by ensuring that enough air is available for the combustion, avoiding overfilling the wood stoves with large logs, avoid leaving the stoves smoldering during nighttime, schedule regular chimney cleaning and professional inspection of the

whole system; (iv) considering meteorological conditions and reduce burning during calmed nights with low ventilation or when haze warnings are issued by weather forecasting or local authorities; and (v) exploring emission control systems such as mechanical separation (altering the velocity or direction of the flue gas through mechanical forces) and electrostatic precipitators (charging the suspended particles in an electric field, to facilitate their removal from the gas flow using electrical forces).

To our knowledge, the results presented in this study, in complement to the findings of Glojek et al. (2022), are the first to show such significant pollution levels and light absorption coefficients from residential wood burning emissions in rural Central Europe. Our findings suggest that the severity and impacts of rural RWB emissions on air quality and climate in the region might be underestimated and overlooked. Additional monitoring and modeling should be considered since the use of RWB is rising worldwide.

**Code and data availability**

The R codes used to curate and analyze the datasets and produce the figures and results of the study are available from the corresponding author upon request. The study data are available at: https://doi.org/10.5281/zenodo.10460257.

**Supplement**

The supplementary material of this study is available online at:

**Author contributions**

AW, GM, LD, MO, and KG conceptualized and designed the study. KG, KW, GM, LD, MO, MM, DvP, and MR set up, operated, calibrated, maintained the instruments during the field study. KG and AG performed the field measurements with support from KW. DvP and HH provided data from the characterization of the $PM_{10}$ filter-based measurements. BR performed

the Mie modeling. ACM curated, processed, and completed the formal data analysis with inputs from KG, TM, GM, AG, and
MR. ACM wrote and prepared the original manuscript draft with inputs from KG, TM, GM, and MP. All the authors reviewed,
edited, and contributed to the scientific discussion in the manuscript.

## Competing interests

The authors declare that they have no conflict of interest.

## Acknowledgments

The authors acknowledge the financial support from the Slovenian Research Agency (program MR-2016, 680 program P1-
0385 "Remote sensing of atmospheric properties"), Municipality of Loški Potok, and the COST Action CA16109
COLOSSAL. We want to thank all the people and institutes involved in the campaign. We are genuinely grateful to the local
community for their friendly welcome, help, and support.

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
