# Peer review of "Optical properties and simple forcing efficiency of the organic aerosols and black carbon emitted by residential wood burning in rural Central Europe"

_EGUsphere, 2023_

## Author Comment (AC1)

**Correspondence to Anonymous Referee #1**

The authors thank the valuable comments and inputs from Anonymous Referee #1. This document contains the responses to each comment. The referee comments are in blue font, while the authors' responses are in black.

This publication highlights the effects of wood heating on air quality, which is important for our health and also for our climate. Experimental data were collected at a rural site where many wood-fired heating systems are in operation during the winter. Special attention will be given to particle properties that are relevant to the radiative forcing of these anthropogenic aerosols.

The work is important to the scientific community because it quantifies the existing particulate air pollution at this site and describes the properties of the particles that will allow

I recommend the paper for publication after the authors address the issues listed below.

One conceptual weakness in the interpreted data is the following:

An important metric used in the paper is PM1, which was calculated from the number-size distributions using density and shape assumptions. However, the size spectra were only recorded up to 600 nm (or 800 nm), and if one looks more closely at e.g. Figure 3f, one clearly has to assume that there is a substantial particle volume between 600 nm (800 nm) and 1000 nm. This does not seem to be taken into account and leads to a significant bias towards too low PM1 values. As a first step, the measured size distributions should be extrapolated into this gap by making appropriate assumptions (log-normal surface distribution or volume distribution?). A discussion and estimation of the resulting errors is mandatory.

Another important problem is that eq 14 is wrong (see below).

Response: Thanks for your observations. The $PM_1$ mass concentrations at the main station (Retje village) were calculated from the particle number size distributions measured by the Tropos homemade MPSS (Tropos Ref. No. 1, Hauke-type medium DMA with a TSI CPC model 377). The instrument covers an actual mobility diameter range from 9.1 to 849.7 nm. In contrast, the PNSD at the background station was measured using a TSI MPSS (TSI Inc., DMA model 3081 with a TSI CPC model 3785), which covers a mobility diameter range from 11.8 to 593.5 nm. Table 1 displays the mobility diameter ranges covered by both instruments, which were erroneously named aerodynamic diameters initially; however, we have corrected the terminology in the table. It is important to highlight that the OA optical properties depending on the OA mass, are reported for the main station (village) only.

By definition, $PM_1$ is composed of particles with an aerodynamic diameter of less than 1 μm. The TROPOS MPSS was designed to cover the $PM_1$ aerodynamic diameter range. From the mathematical relationship between mobility diameter ($d_m$) and aerodynamic diameter ($d_a$) (neglecting the slip correction), we have (Hinds, 1999):

$$d_a = \sqrt{\frac{\rho_p}{\chi * \rho_0}} * d_m, \tag{1}$$

Where $\rho_p$ is the particle density, $\chi$ is the shape factor, and $\rho_0$ is the reference density (1 g cm⁻³). Due to the high load of organic aerosols in the study site, we assumed a particle density of 1.4 g cm⁻³ (Turpin and Lim, 2001). The shape factor was taken as 1 (spherical particles). Replacing these parameters in

equation 1, for an aerodynamic diameter of 1000 nm (PM$_1$), the corresponding mobility diameter is 845.2 nm. The last demonstrates that the mobility diameter covered by the TROPOS MPSS PNSD contains PM$_1$. Therefore, in our judgment, extrapolating the particle number size distribution is unnecessary.

Note: The PNSD measured in the background was used to calculate the differences in aerosol concentrations between the village and the background stations for the three atmospheric stability classes. Deltas in the number concentration ($\Delta$N) were computed using ~600 nm as the upper limit of both size distributions for comparability. This information is indicated in the caption of Figure 4. The calculated $\Delta$OA was excluded from section 3.1 because, only in this case, a PM$_1$ mass concentration from the background station was used in the manuscript. However, considering the maximum mobility diameter covered by the instrument in the background station, it is unsuitable to calculate PM$_1$ and, consequently, OA.

Concerning equation 14, we observed a typo wherein the term $b_{abs}(950\ nm)$ was incorrectly stated instead of the correct notation $b_{abs}(\lambda_1)$. Nevertheless, we have corroborated and can confirm that the correct form of the equation $b_{abs,BrC}(\lambda_1) = b_{abs}(\lambda_1) - b_{abs,BC}(\lambda_1)$ was used in our calculations. The equation has been rectified in the text. Please note that the equation numbering was updated throughout the entire manuscript, and equation 14 is now referred to as equation 12.

**Specific comments (in order)**

- Line 21: "more common in rural areas". Is this true? In some cities, wood is also used for heating and dominates air quality in winter.

Response: Agree. Wood burning is also common in cities and might be responsible for extreme pollution events during winter. However, existing studies need more attention to rural areas where wood burning can be the exclusive or predominant heating source during the colder period. For more clarity, we have rewritten this section in the abstract as follows:

"Existing studies that characterize wood-burning aerosol emissions in Europe primarily concentrate on urban and background sites and focus on BC properties. Despite the significant RWB emissions in rural areas, these locations have received comparatively less attention. The present scenario underscores the imperative for an improved understanding of RWB pollution, aerosol optical properties, and their subsequent connection to climate impacts, particularly in rural areas."

- Lines 40-44: RWB is also important for the health of the local population. This could be mentioned in the abstract. Out of curiosity, has there been an epidemiological study of health effects at the site? Could be informative.

Response: Unfortunately, there were no epidemiological studies on the site during the campaign. Nevertheless, we have included a mention of the risk to human health in the first paragraph of the abstract (line 18).

- Line 95, end of intro: Suggestion to add a few lines pointing out technical solutions to make wood burning cleaner (better certified stoves, appropriate fuel, burning conditions, electrostatic precipitators).

Response: We appreciate the suggestion and have incorporated a series of technical recommendations to mitigate wood-burning emissions in the Summary and Conclusions section, believing this is a more appropriate context to comment on this matter (lines 616 to 626).

- Lines 113-114: The difference in m a.s.l. is 200 m and contradicts the information in Fig. 1.

Response: Thanks for spotting the inconsistency. The main station was located at 715 m a.s.l., while the second station was at 815 m a.s.l.; we have updated Fig. 1.

- Line 163: "Contribution from fibers": Please be more precise. On the one hand directly, but probably also by condensation of semi-volatile gases on the fibers.

Response: Thanks for this observation. In effect, some studies report the condensation of semi-volatile gaseous species on the filter fibers as a source of enhanced light absorption. For instance, Weingartner et al. (2003) observed this effect on filter-based photometers during the first minutes of sample collection on a fresh filter spot after a tape advance. The authors suggest that the filter reaches an equilibrium with the gas phase after a short period, reducing the condensation artifact.

To be more precise, we have rephrased this paragraph as follows:

"Multiple studies point out that the scattering of light in filter-based absorption photometers is affected by the aerosol particles deposited on the filter matrix, their single scattering albedo (SSA), and the scattering within the filter fibers (Ajtai et al., 2019; Bernardoni et al., 2021; Collaud Coen et al., 2010; Drinovec et al., 2022; Saturno et al., 2017; Yus-Díez et al., 2021). Furthermore, some studies suggest that condensation of semi-volatile organic compounds on the filter might contribute to an apparent absorption enhancement (Cappa et al., 2008; Weingartner et al., 2003)."

- Lines 164-169: This is a can of worms and very unsatisfactory. On the one hand, a site-dependent empirical correction for multiple scattering effects in the filter (C) is used, and on the other hand, the harmonization factor is used. Both factors are determined empirically and influence each other. This makes it difficult to compare different instruments on different types of aerosols. A more thorough discussion is needed to disentangle the two factors (C and H). In addition: I also assume that C and H are wavelength dependent - correct? Please clarify.

Response: We acknowledge that using an additional correction factor might be confusing; however, the intention is to correct for previously observed overestimation of the light absorption coefficients calculated from AE33 measurements. In the following paragraphs, we try to provide further details and clarity; also, additional information was incorporated in the revised manuscript in the subsection "The Aethalometer AE33 and multiple scattering harmonization".

The internal algorithm of the AE33 uses the multiple scattering correction factor $C$ to convert the attenuation change $b_{ATN}(\lambda)$ into a light absorption coefficient $b_{abs}(\lambda)$, as follows (Magee Scientific, 2018):

$$b_{abs}(\lambda) = \frac{b_{ATN}(\lambda)}{C}, \tag{2}$$

Equation 2 is a condensed representation of the internal calculation. Here, the value of $C$ is fixed in the instrument, and the instrument manual reports its value depends on the type of filter used during the measurements (Magee Scientific, 2018). Since the release of the AE33, three types of filter tapes have been available: M8020 (also called T60A20), M8050, and M8060. The filters M8020 and M8050 have been discontinued. Currently, users are advised to employ the newest filter, M8060, to minimize the unit-to-unit variabilities associated with the properties of the filter material. The M8060 filter tape has a corresponding $C = 1.39$, which is not site-dependent.

However, the multiple scattering in filter-based absorption photometers is not only filter-dependent since the sampled aerosol particles also contribute to light scattering in the filter fiber depending on their single scattering albedo (Collaud Coen et al., 2010; Saturno et al., 2017; Yus-Díez et al., 2021). To account for the aerosol effect, ACTRIS proposes to include an additional correction factor, $H$ (Müller

and Fiebig, 2021). This "harmonization" factor attempts to harmonize the AE33 measurements with the MAAP measurements, which is treated as the closest to a reference for online aerosol light absorption measurements. The use of an additional correction for multiple scattering is recommended since it has been shown that the default $C$ in the AE33 drives an apparent absorption enhancement resulting in overestimated values of $b_{abs}(\lambda)$ and consequently, BC mass concentrations. This additional correction for absorption enhancement is imperative since the overestimation of absorption adds uncertainty to the calculated aerosol optical properties and simple forcing efficiency. The proposed harmonization factor is calculated as follows:

$$H = \frac{b_{abs}^{AE33,637\,nm}}{b_{abs}^{MAAP,637\,nm}}, \qquad (3)$$

Where, $b_{abs}^{AE33,637\,nm}$ is the light absorption coefficient from the AE33 interpolated to 637 nm (using the measurements at 590 nm and 660 nm), and $b_{abs}^{MAAP,637\,nm}$ is the light absorption coefficient retrieved from the MAAP. ACTRIS has calculated a median value of $H = 1.76$ for the AE33 using the filter tape M8060 based on measurements from multiple European sites, including background and urban stations (Müller and Fiebig, 2021). An AE33-MAAP harmonization factor is now used in recent studies involving AE33 measurements (see for example Pilz et al., 2022; Savadkoohi et al., 2023).

At TROPOS, we have performed parallel measurements using AE33 and MAAP in the laboratory and field campaigns, observing the AE33 measurements to be consistently higher than MAAP. For instance, in Melpitz, Germany, two sets of AE33-MAAP ran in parallel during one year. The data sets show an overestimation of the $b_{abs}$ reported by the AE33 varying from 20 to 50 % (see Figure 1). The median values of $H$ calculated for Melpitz during the one-year campaign were 2.25 (ranging between 1.74 (P.10) and 2.97 (P.90)) at Melpitz research station, and 1.87 (ranging between 1.15 (P.10) and 2.83 (P.90)) at Melpitz village station.

[Figure]

Figure 1: Time series of light absorption coefficient measurements from the AE33 and MAAP at (a) Melpitz research station and (b) Melpitz village station.

Due the absence of concurrent AE33-MAAP measurements in Loški Potok, we employed a value of $H$ = 1.9 obtained from measurements at Melpitz village during wintertime. Melpitz is a small village (~200 inhabitants) located in eastern Germany at 50 km from Leipzig. During the campaign, a measurement container was placed in the center of the village where residential wood burning is the main heating source during the coldest season (van Pinxteren et al., 2023). Although the $H$ value for Melpitz-winter is slightly higher than the $H$ recommended by ACTRIS, the difference between both is small, only 7 %.

The light absorption coefficients in the study site were harmonized as shown below:

$$b_{abs}(\lambda)^{Harm.} = \frac{b_{ATN}(\lambda)}{C*H} = \frac{b_{abs}(\lambda)}{H} \tag{4}$$

In our study, $C$ and $H$ are taken as wavelength-independent parameters. The spectral dependency of the light scattering corrections in the aethalometers has been investigated in a few studies with divergent results. For the AE33, Bernardoni et al. (2021) found no statistically significant change in $C$ with wavelength. Yus-Díez et al. (2021) investigated the effective multiple scattering parameter ($C$) defined as a function of the cross-sensitivity to scattering, the multiple scattering filter parameter, and the aerosol SSA. In three diverse environments, the authors observed no wavelength-dependency of $C$ on the urban and regional stations, while they found variability in a mountaintop station. The variability in the remote mountain site was explained by the aerosols SSA: samples with low SSA (< 0.9) did not show a wavelength dependency in $C$, while the opposite was found in samples with high SSA (> 0.9). In Melpitz, Tropos measurements have shown that in winter, SSA < 0.9. For Loški Potok, the SSA estimated through Mie modeling did not exceed 0.9 (see Table 1).

**Table 1.** Aerosol SSA calculated for Loški Potok, Slovenia, using Mie modeling

| Wavelength | Mean | P.10 | P.90 |
|---|---|---|---|
| 370 | 0.47 | 0.42 | 0.55 |
| 470 | 0.49 | 0.41 | 0.59 |
| 520 | 0.53 | 0.44 | 0.63 |
| 590 | 0.56 | 0.47 | 0.65 |
| 660 | 0.61 | 0.53 | 0.7 |
| 880 | 0.65 | 0.57 | 0.73 |

Based on the previous findings and considering the mathematical definition of $H$, we have assumed that $C$ and $H$ do not exhibit spectral variation. Nevertheless, further investigation into this matter is needed but is beyond the scope of our study.

- Line 180, Table 1: What is the weighing procedure for PM10? The table is not complete. I am missing information on offline TCA, ion chromatography, levoglucosan. Was an impactor used in the SMPS that allows the correct correction for multiple charging? This is essential for correct volume determination. Why does one SMPS measure only up to 600 nm and the other up to 800 nm? However, the aerodynamic diameter is given. How was this converted?

Response: A detailed description of the instrumentation and analytical methods used in the study is given in section 2.2 of the manuscript. We have included further details in this section to complement the missing information. Below, we answer each point addressed in this comment.

Filter samples of particulate matter $PM_{10}$ were collected every 12 hours at the village station (06:00 to 18:00 and 18:00 to 06:00, local time) using a high-volume sampler (DHA-80, Digitel). $PM_{10}$ mass concentrations were determined gravimetrically in the laboratory following the European Union standard EN 12341. The quartz fiber filters (150 mm diameter) were preheated before sampling for at least 24 hours at 105 °C to minimize blank values and frozen after sampling until their characterization in the laboratory; the weighing was done using a microbalance (AT261 Delta Range, Mettler-Toledo). The $PM_{10}$ filters were analyzed to estimate the particle's chemical composition, including organic and elemental carbon (OC/EC), ions ($NH4+$, $Cl^-$, $Na^+$, $K^+$, $Mg^{2+}$, $Ca^{2+}$, $NO^{3-}$, $SO_4^{2-}$, and $C_2O_4^{2-}$), and levoglucosan. Mass concentrations of OC/EC were quantified following the EUSAAR-2 Protocol (Cavalli et al., 2010); ions were estimated using ion chromatography (Dionex ICS3000) of ultrapure water extracts (further details in Fomba et al., 2014), and levoglucosan was quantified using high-performance anion exchange chromatography coupled with an electrochemical detector (HPAEC-PAD, Iinuma et al., 2009).

The total carbon (TC) mass concentration at the village station was measured using an online Total Carbon Analyzer (TCA08, Magee Scientific). The dual-chamber instrument uses an online thermal method to quantify the total carbonaceous fraction in atmospheric aerosols. In one chamber, the sample is collected over a quartz fiber filter (47 mm diameter) and heated to 940 °C, transforming the carbon compounds into $CO_2$. The amount of $CO_2$ is measured before and after combustion by a $CO_2$ detector and later integrated to calculate the total carbon mass concentration. In parallel, the second chamber collects a new aerosol sample (sampling time adjustable from 20 min to 24 h). Both chambers alternate between sampling and analysis, enabling online functionality. Further details about the TCA are given in Rigler et al. (2020).

The Tropos MPSS did not employ pre-impactors. The multiple charge correction was done using a multiple-charge inversion routine to the raw mobility distributions described in Pfeifer et al. (2014). This information is mentioned in section 2.2.

The difference in the particle sizes covered the background MPSS (TSI Inc., DMA model 3081 with a TSI CPC model 3785) and the main-station MPSS (TROPOS Ref. No. 1, Hauke-type medium DMA with a TSI CPC model 3772) resides in the distinct high-voltage power supplies and the geometries of the DMA. The TSI MPSS uses a 10 kV high-voltage power supply and a TSI design DMA, which allows the instrument to reach a maximum particle size of ~600 nm. In contrast, the TROPOS MPSS uses a 12.5 kV high-voltage power supply, including a homemade TROPOS design DMA (Hauke-type).

Regarding the aerodynamic diameters mentioned in Table 1, we have corrected them to indicate that these are mobility diameters.

- Line 182, eq 2: why does it say "fraction in PM"? this is misleading because a fraction is unitless - the rest of the equation is not unitless...

Response: Many thanks for the observation; we have corrected the subscript to "PM."

- Line 188: Suggest writing PM0.8 or PM0.6 instead of PM1.

Response: Considering our answer to the first comment of this review, we opted to maintain the terminology as $PM_1$.

- Line 210, Figure: I expect large systematic errors affecting the slope. Please discuss.

Response: The systematic error of the slope in Fig. 2 was calculated using error propagation, considering the contribution of individual uncertainties involved in the determination of $PM_1$ (sampling

flow rate, volumetric diameter, and particle density) and $PM_{10}$ (air sampling volume, reproducibility). The individual uncertainties are 17 % for $PM_1$ and 10 % for $PM_{10}$ (Hafkenscheid, 2013; Leiva et al., 2012). For a slope of 0.9, the resulting uncertainty is 15%. This is now described in the text (see lines 229 to 231). Also, further details have been included in section 3.3.

- Line 214: Is OA_MPSS = PM1? please be consistent and use the same names.

Response: In this paragraph, $OA_{MPSS}$ refers to the organic aerosol mass concentration calculated from the mass balance $OA = PM_1 - eBC - InA$. This enables the differentiation with the other estimated OA (or OC) mass concentrations, i.e., from the $PM_{10}$ filters ($OC_{filters}$) and the total carbon analyzer ($OA_{TCA}$).

In the rest of the document, $OA_{MPSS}$ is referred to as OA; we have indicated this in line 245.

- Line 222: Is it justified to assume that transmission, albedo, backscatter fraction are constant, i.e. not wavelength dependent? For which part of the electromagnetic spectrum is this true?

Response: In our calculation of the simple radiative forcing efficiency, the atmospheric transmission ($\tau_{atm}$) and surface albedo ($a_s$) were assumed wavelength-independent. The backscatter fraction ($\beta$) was calculated as wavelength-dependent, as described further below.

The atmospheric transmission can be assumed wavelength-independent in certain spectral regions where atmospheric constituents such as gases and aerosols have minimal absorption or scattering. This is true for given bands; for instance, water vapor has a minimal absorption effect in some bands in the near-IR and IR regions. However, this assumption is not valid in other regions of the electromagnetic spectrum, such as the ultraviolet ($\lambda \leq 320$ nm) or certain infrared bands where the atmospheric transmission can be highly wavelength-dependent. In summary, the wavelength dependence or independence of $\tau_{atm}$ varies along the electromagnetic spectrum.

In our study, we have assumed a wavelength-independent $\tau_{atm}$ since we intend to present a sensitivity analysis to represent the effect of including or neglecting the OA optical properties calculated from our filed measurements. Additionally, using constant values of $\tau_{atm}$ and $a_s$ allow us to compare our results with other studies using a similar forcing efficiency model (e.g. Chen and Bond, 2010; Deng et al., 2022). The value of $\tau_{atm}$ attempts to be representative of the variability given in the visible spectra and corresponds to the geometric mean of the average downward transmission (0.72) and the reflected upward transmission (0.87), as proposed by Chylek and Wong (1995) and Penner et al. (1992).

The surface albedo ($a_s$) can be, in general, assumed as wavelength-independent when the properties of the surface do not change along the electromagnetic spectrum, i.e., the scattered light by the surface is the same or changes relatively little with the wavelength. This assumption is valid for natural surfaces like soil, deserts, and vegetation. Nevertheless, for vegetation, the surface albedo is primarily low in the visible spectrum and comparatively higher in the IR; it changes depending on the vegetation type and season. For snow, the albedo is strongly dependent on the surface conditions (age, cleanness, roughness) and exhibits a wavelength dependency. In the spectral range from 500 to 800 nm, the albedo of fresh snow is nearly constant (~0.8 for ice-crustal snow, ~0.7 for large-grained wet snow) and decreases on the extremes of that range (0.5 to 0.8) (Iqbal, 1983). In our study, we used average albedos for fresh snow (0.8) and Earth average (0.19), representing diverse surface features and conditions. For the spectral range covered in our forcing efficiency calculations (370–880 nm), the approximations of the spectral dependence of albedo are frequently made, and we assume that the reflectivity of the surface remains reasonably constant. This assumption also enables comparability with other studies, such as Chen and Bond (2010).

The backscatter fraction $\beta(\lambda)$ was taken as wavelength dependent and estimated by the mathematical relation with the asymmetry parameter $g(\lambda)$ proposed by Sagan and Pollack (1967):

$$\beta(\lambda) = \frac{1}{2}(1 - g(\lambda))$$

The asymmetry parameter was modeled using core-shell Mie theory simulations. This information is described in section 2.4 of the manuscript.

- Line 292: Is it EC or BC (EC is present with lower time resolution)?

Response: In this section, we refer to EC mass concentrations estimated from thermo-optical methods on the $PM_{10}$ filters. The resolution is lower than BC since the sampling of the $PM_{10}$ filters was done every 12 hours.

- Line 300, Fig 3: the color for "unstable" is hard to distinguish, perhaps better in green. Fig 3b: At what wavelength was BC measured? Fig 3f: here the spectra go up to 850 nm. An additional plot of the volume size distribution would be helpful.

Response: Thanks for this observation. We have modified Fig. 3 using a different color palette. BC in Fig. 3b was calculated using absorption measured at 950 nm. The higher diameter in the size distribution measured by the Tropos MPSS is 849 nm.

- Line 327, caption Fig. 4: the black points (outliers) are not visible.

Response: Thanks for this observation; we have corrected the caption in Fig. 4.

- Line 332, Fig. 5: ditto

Response: Corrected.

- Line 335 and elsewhere: given the relatively large uncertainties, it makes no sense to give the values so precisely. Here 1100 mM-1 would be appropriate.

Response: Following the recommendation, we have rounded the figures to two significant digits.

- Line 347, eq 12: This relationship is general and one could remove the BC here.

Response: We opted to keep BC in the equation (now Eq. 10) since the apportionment method requires two assumptions: (i) $AAE_{BC} = 1$, and (ii) the total light absorption coefficient in the near IR is entirely attributed to BC.

- Line 351, eq 13: The exponent is an equation and therefore misleading. Just write -1 as the exponent.

Response: Agree. Equation 13 (now referred to as Eq. 11) has been corrected.

- Line 335, eq 14: I think this equation is clearly wrong. It should be: b_abs.BrC(l1)=b_abs(l1)-b_abs(950)*(l1/950)^-1.

Response: Thanks for this important observation. Equation 14 (now referred to as Eq. 12) had a typo; the term $b_{abs}(950\ nm)$ was erroneously stated instead of the correct notation $b_{abs}(\lambda_1)$. Yet, we have corroborated and can confirm that the correct form of the equation $b_{abs,BrC}(\lambda_1) = b_{abs}(\lambda_1) - b_{abs,BC}(\lambda_1)$ was used in our calculations.

- Line 366, Fig. 6: Will this figure change if eq. 14 is changed? Depending on this, it will also lead to an adjustment of the discussion (e.g. lines 369-377).

Response: Given that the correct equation was used in our calculations (see previous answer), the apportioned light absorption coefficients do not change, and neither do the contributions of BC and BrC shown in Fig. 6.

- Line 394: Will the "photochemical process" lead to an increase or decrease of AAE_BrC?

Response: In our study, the $AAE_{BrC, 370-590 nm}$ decreased with increased atmospheric instability, which might be related to the mixing with external aerosol sources and aerosol aging/photochemical processes occurring during unstable conditions. The median $AAE_{BrC, 370-590 nm}$ were 3.9, 3.7 and 3.5 for strong inversion, weak inversion, and unstable atmosphere, suggesting a reduction in $AAE_{BrC, 370-590 nm}$ with increased photochemistry. In the literature, some studies suggest a decrease in light absorption in the near UV after aging and photobleaching. For instance, Deng et al., (2022) found comparatively lower BrC wavelength dependency during summer, and suggest that photobleaching and aerosol aging involved in the formation of secondary organic aerosol (SOA) might explain the reduced optical properties (including AAE), during summer time. In other study, Rana et al. (2020) found a reduction of at least 50 % in $MAC_{BrC}$ after aging and transport of highly loaded BrC plumes in the Indo-Gangetic Plan. The lower MAC suggests a reduction in the light absorption at shorter wavelengths, reducing the wavelength dependency of BrC.

- Lines 403-420: Again, the problem with PM1: How much does the missing volume affect the MAC values? I would like to see a presentation and discussion of the systematic errors.

Response: We have included uncertainty calculations in the revised manuscript using error propagation. To estimate the $MAC_{OA}$ uncertainty, we considered the individual contributions of the OA mass concentration and BrC light absorption coefficients. The propagated error of the OA mass includes the uncertainty derived from the $PM_1$ mass calculation and accounts for contributions from the sampling flow rate, the volumetric diameter, the diffusion efficiency corrections, and the particle density. The $PM_1$ uncertainty was assumed to be 17 % based on the calculations of Buonanno et al. (2009) and the instrument intercomparison in the laboratory. Furthermore, the contribution of the individual uncertainties of the eBC and inorganic species mass concentrations were included in the calculations. The resulting uncertainty for the $MAC_{OA, 370 nm}$ is 46 %. An extended description of the uncertainty estimation is now given in the new section 3.3 of the revised manuscript.

- Line 429: Regarding the measurement conditions: How were the particles sampled to the instruments (sampling conditions, at what temperatures and thus relative humidities were the particles measured)?

Response: Nafion®Permapure air dryers (length=1.5 m) were used to keep the relative humidity of the AE33 and MPSS samples below 40 %. The instruments operated under ambient room temperature. For $PM_{10}$, the filters were preheated before sampling for at least 24 hours at 105 °C to minimize blank values and frozen after sampling until their characterization in the laboratory. The conditions of sampling and analysis are described in section 2.2 of the manuscript (see lines 155 to 158).

- Line 450: The beta should be a_s.

Response: Many thanks for noticing this error; we have corrected it.

- Line 453: I have recalculated the RF values in Fig. 9 graphically and get about 20% lower values. Please check the integration. Note that in Fig. 9 the wavelengths are not equidistant as shown!

Response: We have corroborated manually and in R our calculations to obtain RF and found similar values in both cases. The RF is calculated as the area under the curve formed by the median SFE at each wavelength, applying the trapezoidal rule for uneven intervals. The results are still the same as those reported in the manuscript. We have adjusted Fig. 9 to show the accurate distances among the wavelengths in the x axis.

- Line 460,461: two times: inverse square meter

Response: The units are given as W m$^2$ nm$^{-1}$.

- Line 466: the lensing effect was not described before. Have you compared the MAC_BC with literature values? Should it be higher in this study?

Response: Thanks for the observation. The lensing effect was briefly mentioned in the introduction; however, further details have been included in the revised manuscript (see lines 76 to 77). Typical $MAC_{BC}$ values fall in the range of 5 to 15 m$^2$ g$^{-1}$ for wavelengths between 500 and 880 nm and exhibit an inverse relationship with wavelength (Bond and Bergstrom, 2006; Feng et al., 2021; Mbengue et al., 2021). At 550 nm, Bond and Bergstrom (2006) suggest that BC has a reliable MAC of 7.5 ± 1.2 m$^2$ g$^{-1}$. Tropos' measurements at the village of Melpitz yielded a $MAC_{BC, 880\,nm}$ of 6.5 m$^2$ g$^{-1}$, using harmonized light absorption coefficients. The calculated $MAC_{BC, 950}$ nm for Loški Potok, was 6.7 m$^2$ g$^{-1}$, which remains within the range of previously reported values and is similar to $MAC_{BC}$ for Melpitz. For Loški Potok, the $MAC_{BC}$ could be comparatively lower than other studies because of the high wavelength for which it is reported and the introduction of the harmonization of the light absorption coefficients. We know a fraction of previous studies has not considered additional corrections for the aerosol scattering contribution on filter-based absorption photometers.

- Line 470, Table 3: The min and max values are not very meaningful because they depend on the choice of the averaging interval. Better would be e.g. quantiles

Response: We respectfully prefer to maintain min and max values; however, we have included percentiles 10 and 90 in Tables 3 and 4.

- Line 489, 490: Consider (again) the number of significant digits. Put the units after the whole expression: e.g: 71 +- 56 ug/m3.

Response: We have adjusted our figures to keep two significant digits. Also, we have changed the position of the units.

References

Ajtai, T., Kiss-Albert, G., Utry, N., Tóth, Á., Hoffer, A., Szabó, G. and Bozóki, Z.: Diurnal variation of aethalometer correction factors and optical absorption assessment of nucleation events using multi-wavelength photoacoustic spectroscopy, J. Environ. Sci. (China), 83, 96–109, doi:10.1016/j.jes.2019.01.022, 2019.

Bernardoni, V., Ferrero, L., Bolzacchini, E., Corina Forello, A., Gregorič, A., Massabò, D., Mocnik, G., Prati, P., Rigler, M., Santagostini, L., Soldan, F., Valentini, S., Valli, G. and Vecchi, R.: Determination of Aethalometer multiple-scattering enhancement parameters and impact on source apportionment during the winter 2017/18 EMEP/ACTRIS/COLOSSAL campaign in Milan, Atmos. Meas. Tech., 14(4), 2919–2940, doi:10.5194/amt-14-2919-2021, 2021.

Bond, T. C. and Bergstrom, R. W.: Light absorption by carbonaceous particles: An investigative review, Aerosol Sci. Technol., 40(1), 27–67, doi:10.1080/02786820500421521, 2006.

Buonanno, G., Dell'Isola, M., Stabile, L. and Viola, A.: Uncertainty budget of the SMPS-APS system in the measurement of PM 1, PM2.5, and PM10, Aerosol Sci. Technol., 43(11), 1130–1141, doi:10.1080/02786820903204078, 2009.

Cappa, C. D., Lack, D. A., Burkholder, J. B. and Ravishankara, A. R.: Bias in filter-based aerosol light absorption measurements due to organic aerosol loading: Evidence from laboratory measurements, Aerosol Sci. Technol., 42(12), 1022–1032, doi:10.1080/02786820802389285, 2008.

Cavalli, F., Viana, M., Yttri, K. E., Genberg, J. and Putaud, J.-P.: Toward a standardised thermal-optical protocol for measuring atmospheric organic and elemental carbon: the EUSAAR protocol, Atmos. Meas. Tech., 3(1), 79–89, doi:10.5194/amt-3-79-2010, 2010.

Chen, Y. and Bond, T. C.: Light absorption by organic carbon from wood combustion, Atmos. Chem. Phys., 10(4), 1773–1787, doi:10.5194/acp-10-1773-2010, 2010.

Chylek, P. and Wong, J.: Effect of absorbing aerosols on global radiation budget, Geophys. Res. Lett., 22(8), 929–931, doi:10.1029/95GL00800, 1995.

Collaud Coen, M., Weingartner, E., Apituley, A., Ceburnis, D., Fierz-Schmidhauser, R., Flentje, H., Henzing, J. S., Jennings, S. G., Moerman, M., Petzold, A., Schmid, O. and Baltensperger, U.: Minimizing light absorption measurement artifacts of the Aethalometer: Evaluation of five correction algorithms, Atmos. Meas. Tech., 3(2), 457–474, doi:10.5194/amt-3-457-2010, 2010.

Deng, J., Ma, H., Wang, X., Zhong, S., Zhang, Z., Zhu, J., Fan, Y., Hu, W., Wu, L., Li, X., Ren, L., Pavuluri, C. M., Pan, X., Sun, Y., Wang, Z., Kawamura, K. and Fu, P.: Measurement report: Optical properties and sources of water-soluble brown carbon in Tianjin, North China - insights from organic molecular compositions, Atmos. Chem. Phys., 22(10), 6449–6470, doi:10.5194/acp-22-6449-2022, 2022.

Drinovec, L., Jagodič, U., Pirker, L., Škarabot, M., Kurtjak, M., Vidović, K., Ferrero, L., Visser, B., Röhrbein, J., Weingartner, E., Kalbermatter, D. M., Vasilatou, K., Bühlmann, T., Pascale, C., Müller, T., Wiedensohler, A. and Močnik, G.: A dual-wavelength photothermal aerosol absorption monitor: design, calibration and performance, Atmos. Meas. Tech., 15(12), 3805–3825, doi:10.5194/amt-15-3805-2022, 2022.

Feng, X., Wang, J., Teng, S., Xu, X., Zhu, B., Wang, J., Zhu, X., Yurkin, M. A. and Liu, C.: Can light absorption of black carbon still be enhanced by mixing with absorbing materials?, Atmos. Environ., 253(February), 118358, doi:10.1016/j.atmosenv.2021.118358, 2021.

Fomba, K. W., Müller, K., Van Pinxteren, D., Poulain, L., Van Pinxteren, M. and Herrmann, H.: Long-term chemical characterization of tropical and marine aerosols at the Cape Verde Atmospheric Observatory (CVAO) from 2007 to 2011, Atmos. Chem. Phys., 14(17), 8883–8904, doi:10.5194/acp-14-8883-2014, 2014.

Hafkenscheid, T. L.: Inter-laboratory comparison of the determination of PM10 in ambient air using filter sampling and weighing., 2013.

Hinds, W. C.: Aerosol Technology: Properties, Behavior, and Measurement of Airborne Particles, 2nd editio., John Wiley & Sons, New York., 1999.

Iinuma, Y., Engling, G., Puxbaum, H. and Herrmann, H.: A highly resolved anion-exchange chromatographic method for determination of saccharidic tracers for biomass combustion and primary bio-particles in atmospheric aerosol, Atmos. Environ., 43(6), 1367–1371, doi:10.1016/j.atmosenv.2008.11.020, 2009.

Iqbal, M.: Ground Albedo, in An Introduction to Solar Radiation, pp. 281–293., 1983.

Leiva, G. M. A., Araya, M. C., Alvarado, A. M. and Seguel, R. J.: Uncertainty estimation of anions and cations measured by ion chromatography in fine urban ambient particles (PM 2.5), Accredit. Qual. Assur., 17(1), 53–63, doi:10.1007/s00769-011-0844-4, 2012.

Magee Scientific: Aethalometer® Model AE33 User Manual, , (ver. 1.57), 149 [online] Available from: www.aerosol.eu, 2018.

Mbengue, S., Zikova, N., Schwarz, J., Vodička, P., Šmejkalová, A. H. and Holoubek, I.: Mass absorption cross-section and absorption enhancement from long term black and elemental carbon measurements: A rural background station in Central Europe, Sci. Total Environ., 794, doi:10.1016/j.scitotenv.2021.148365, 2021.

Müller, T. and Fiebig, M.: ACTRIS In Situ Aerosol: Guidelines for Manual QC of AE33 absorption photometer data. [online] Available from: https://www.actris-ecac.eu/particle-light-absorption.html,

2021.

Penner, J. E., Dickinson, R. E. and O'Neill, C. A.: Effects of aerosol from biomass burning on the global radiation budget, Science (80-. )., 256(5062), 1432–1434, doi:10.1126/science.256.5062.1432, 1992.

Pfeifer, S., Birmili, W., Schladitz, A., Müller, T., Nowak, A. and Wiedensohler, A.: A fast and easy-to-implement inversion algorithm for mobility particle size spectrometers considering particle number size distribution information outside of the detection range, Atmos. Meas. Tech., 7(1), 95–105, doi:10.5194/amt-7-95-2014, 2014.

Pilz, C., Düsing, S., Wehner, B., Müller, T., Siebert, H., Voigtländer, J. and Lonardi, M.: CAMP: An instrumented platform for balloon-borne aerosol particle studies in the lower atmosphere, Atmos. Meas. Tech., 15(23), 6889–6905, doi:10.5194/amt-15-6889-2022, 2022.

van Pinxteren, D., Engelhardt, V., Mothes, F., Poulain, L., Fomba, K. W., Spindler, G., Cuesta-Mosquera, A., Tuch, T., Müller, T., Wiedensohler, A., Löschau, G., Bastian, S. and Herrmann, H.: Residential Wood Combustion in Germany: A Twin-Site Study of Local Village Contributions to Particulate Pollutants and Their Potential Health Effects, ACS Environ. Au, 0(0), null, doi:10.1021/acsenvironau.3c00035, 2023.

Rana, A., Dey, S., Rawat, P., Mukherjee, A., Mao, J., Jia, S., Khillare, P. S., Yadav, A. K. and Sarkar, S.: Optical properties of aerosol brown carbon (BrC) in the eastern Indo-Gangetic Plain, Sci. Total Environ., 716, 137102, doi:10.1016/j.scitotenv.2020.137102, 2020.

Rigler, M., Drinovec, L., Lavri, G., Vlachou, A., Prevot, A. S. H., Luc Jaffrezo, J., Stavroulas, I., Sciare, J., Burger, J., Kranjc, I., Turšič, J., D. A. Hansen, A. and Mocnik, G.: The new instrument using a TC-BC (total carbon-black carbon) method for the online measurement of carbonaceous aerosols, Atmos. Meas. Tech., 13(8), 4333–4351, doi:10.5194/amt-13-4333-2020, 2020.

Sagan, C. and Pollack, J. B.: Anisotropic nonconservative scattering and the clouds of Venus, J. Geophys. Res., 72(2), 469–477, doi:https://doi.org/10.1029/JZ072i002p00469, 1967.

Saturno, J., Pöhlker, C., Massabò, D., Brito, J., Carbone, S., Cheng, Y., Chi, X., Ditas, F., Hrab De Angelis, I., Morán-Zuloaga, D., Pöhlker, M. L., Rizzo, L. V., Walter, D., Wang, Q., Artaxo, P., Prati, P. and Andreae, M. O.: Comparison of different Aethalometer correction schemes and a reference multi-wavelength absorption technique for ambient aerosol data, Atmos. Meas. Tech., 10(8), 2837–2850, doi:10.5194/amt-10-2837-2017, 2017.

Savadkoohi, M., Pandolfi, M., Reche, C., Niemi, J. V., Mooibroek, D., Titos, G., Green, D. C., Tremper, A. H., Hueglin, C., Liakakou, E., Mihalopoulos, N., Stavroulas, I., Artiñano, B., Coz, E., Alados-Arboledas, L., Beddows, D., Riffault, V., De Brito, J. F., Bastian, S., Baudic, A., Colombi, C., Costabile, F., Chazeau, B., Marchand, N., Gómez-Amo, J. L., Estellés, V., Matos, V., van der Gaag, E., Gille, G., Luoma, K., Manninen, H. E., Norman, M., Silvergren, S., Petit, J. E., Putaud, J. P., Rattigan, O. V., Timonen, H., Tuch, T., Merkel, M., Weinhold, K., Vratolis, S., Vasilescu, J., Favez, O., Harrison, R. M., Laj, P., Wiedensohler, A., Hopke, P. K., Petäjä, T., Alastuey, A. and Querol, X.: The variability of mass concentrations and source apportionment analysis of equivalent black carbon across urban Europe, Environ. Int., 178(June), doi:10.1016/j.envint.2023.108081, 2023.

Turpin, B. J. and Lim, H. J.: Species contributions to pm2.5 mass concentrations: Revisiting common assumptions for estimating organic mass, Aerosol Sci. Technol., 35(1), 602–610, doi:10.1080/02786820119445, 2001.

Weingartner, E., Saathoff, H., Schnaiter, M., Streit, N., Bitnar, B. and Baltensperger, U.: Absorption of light by soot particles: Determination of the absorption coefficient by means of aethalometers, J. Aerosol Sci., 34(10), 1445–1463, doi:10.1016/S0021-8502(03)00359-8, 2003.

Yus-Díez, J., Bernardoni, V., Močnik, G., Alastuey, A., Ciniglia, D., Ivančič, M., Querol, X., Perez, N., Reche, C., Rigler, M., Vecchi, R., Valentini, S. and Pandolfi, M.: Determination of the multiple-scattering correction factor and its cross-sensitivity to scattering and wavelength dependence for different AE33 Aethalometer filter tapes: a multi-instrumental approach, Atmos. Meas. Tech., 14(10), 6335–6355, doi:10.5194/amt-14-6335-2021, 2021.

---

## Author Comment (AC2)

**Correspondence to Anonymous Referee #2**

The authors express their gratitude for the valuable feedback provided by Anonymous Referee #2. This document includes responses to each comment, with the referee's comments presented in blue font and the authors' responses in black.

Biomass burning emission is a hot topic in Air Quality in Europe. Biomass burning, mainly related to residential heating has recently increase due to the incentives to reduce greenhouse emissions. These emissions can be very important in medium size cities and in rural areas, and may have impact on both health and climate. As shown in the present article, this can be of great interest in rural areas frequently affected by thermal inversions. Moreover, there is growing interest in evaluating the optical properties of carbonaceous aerosols emitted by biomass combustion. The manuscript corroborated the importance of this source at rural areas and demonstrates the influence of coating of BC by OA in absorption and therefore on atmospheric warming.

This is a 2 months period campaign carried out in the village of Retje, in Slovenia. A complete set of instrumentation was settled at the village and at a reference location, 150 m higher. Instruments comprised: Aethalometers, MPSS, and CPC. Ions and EC/OC were determined at filters collected by high volume samplers. At the village site a total carbon analyzer was also used.

The paper is of interest and deserves to be published in ACP although there are some aspects that can be improved, mainly related to the uncertainty in the estimation of OA.

As stated in the manuscript, estimating OA hourly concentrations by subtracting BC and ions (measured in $PM_{2.5}$ and $PM_{10}$, respectively) from the $PM_1$ mass calculated form MPSS could be the largest source of uncertainty: 1) by the different sizes measured / sampled; 2) the MPSS in Retje measured from 10-800 nm; 3) because the ions were offline estimated in $PM_{10}$ filters collected every 12h and a constant contribution of ions to $PM_1$ has been assumed, affecting the time variation of OA. Ions and EC/OC mainly concentrates in $PM_1$ but presence in the coarser fraction cannot be discarded. It is true that there is a very good correlation between PM10 and PM1 derived from MPSS, indicating 90% of $PM_{10}$ is in the $PM_1$ fraction as an average; but in some cases, with high PM concentrations, $PM_1$ accounts for around 70% of $PM_{10}$ and then there is an important contribution of coarse PM that will affect the OA estimation. The authors compared $OA_{MPSS}$ and $OA_{TCA}$ and concluded that the good correlation corroborates the adequacy of the method used. However, it must be considered that, in both cases, OA/OC ratios used have been estimated by comparing the OA estimated from MPSS with the OC of filters. Therefore, the good correlation between $OA_{MPSS}$ and $OA_{TCA}$ only demonstrates a good correlation between OCtca and OC filters, but does not provide evidence on the suitability of the method used for estimating OA.

This uncertainty in the estimation of OA may have a high impact on the results and conclusions. Thus, it will influence the estimation of $MAC_{OA}$. Then, I considered that more info about OA uncertainty should be provided.

Response: Thanks for your important observation regarding the OA mass concentrations. We recognize that the methodology employed in the OA calculation implies specific assumptions that are sources of uncertainty in the aerosol properties reported, like $MAC_{OA}$. To address this point, we have incorporated uncertainty calculations in the manuscript using error propagation. For the OA uncertainty, the individual sources of error contain the calculation of $PM_1$ mass concentrations from the PNSD, which include the MPSS deviations and aerosol density, the eBC uncertainty, and the contribution from the determination of inorganic aerosol mass concentrations in $PM_1$ (scaling fraction from $PM_{10}$ to $PM_1$, and uncertainties from the analytical method).

The PM$_1$ uncertainty is taken as 17 % based on the calculations of Buonanno et al. (2009) and the MPSS intercomparison in the laboratory. This estimation accounts for contributions from the sampling flow rate, the volumetric diameter, the diffusion efficiency corrections, and the particle density. The uncertainty of the eBC mass concentration is assumed to be 5 %, corresponding to the EC uncertainty, as eBC was normalized to EC. The uncertainty of EC corresponds to the reproducibility of the thermo-optical analysis following the EUSAAR protocol. The uncertainty contribution of the insoluble inorganics was assumed to be 20 %, accounting for the PM$_{10}$-PM$_1$ scaling and repeatability, an important contributor to uncertainty in inorganic content determination, as shown in Leiva et al. (2012). The final uncertainty for the OA mass concentration was 22%. Additional details are given in the new section 3.3.

Regarding the comparison between OA$_{MPSS}$ and OA$_{TCA}$, we agree that the use of a local OA/OC ratio results, in the end, in a biased comparison. Consequently, we have opted to show a comparison between OA$_{MPSS}$ and OC from TCA (OC$_{TCA}$) and filters (OC$_{filters}$) (see updated Fig. S1). The agreements in both comparisons and the resulting OA/OC ratios (slopes) show that the estimated OA$_{MPSS}$ follows a realistic trend of the organic aerosols in the study site and suggest that the hourly OA$_{MPSS}$ are comparable to independent methods. The differences in the estimated OA/OC ratios are related to the distinct sampling periods (1-hour averaged data from TCA and 12-hour data from filters) and techniques (Brown et al., 2013). However, both OA/OC ratios fall within the range of reported results in previous studies (Srinivas and Sarin, 2014; Xing et al., 2013). These changes were included in section 2.3 (see lines 238 to 244) and Fig. S1 in the Supplement.

[Figure]

**Figure S1. Scatter plots and orthogonal regressions (solid black lines) for the comparison of (a) OAMPSS and OCTCA, and (b) OAMPSS and OCfilters.**

Minor corrections

- Line 139. Add reference for TCA

Response: The reference to the TCA was included in the text (see line 148). Additional information about the instrument's principles of operation was also included in section 2.2.

- Line 180. This Table can go to Supplementary. Information on inlets size cut should be added

Response: We appreciate this suggestion; however, we respectfully prefer to keep Table 1 in the main text as a summary of the instrumentation used during the campaign. Nevertheless, information about the inlets is included in section 2.2 (see lines 155 to 158) and is also mentioned in Table 1 now.

- Line 270: Table 2. Can you add the % of hours for each category during the sampling period? Or just shortly describing the frequency of the stability categories in the text.

Response: The percentages of occurrence for each condition of atmospheric stability are included in Tables 3 (hourly basis) and 4 (predominant during periods of 12 hours). A mention on the frequencies of each category is given in lines 334 to 335. For hourly measurements, the percentages of occurrence were 28 % for strong inversion, 31 % for weak inversion, 5 % for neutral atmosphere, and 36 % for unstable atmosphere.

- Line 282 (and Fig.3): Does OA refers to OAmpss? It should be clearly stated that OA refers to OAmpss in the manuscript.

Response: Thanks for this observation. In effect, the term OA refers to $OA_{MPSS}$. We have indicated that $OA_{MPSS}$ is called OA in the manuscript (see line 245).

- Lines 287-289: PNC is very similar for strong inversion and unstable atmosphere.

Response: In this paragraph, we compared the maximum particle number for both categories of atmospheric stability. However, the mean concentrations are significantly different: for strong inversion, the mean N was $17 \times 10^3$ particles $cm^{-3}$, while for unstable atmosphere, the mean N was $6.1 \times 10^3$ particles $cm^{-3}$. We have modified the text and reported mean values of N in section 3.1 (see lines 317 to 318).

- Line 312-317. Little discussion about ΔPNC and PNSD; I understand this is not the topic of the articles. PNC measurements have been mainly used for deriving PM1 and hourly OA. However, I would add an explanation about similarity of ΔPNC for N10-50 during the three categories

Response: We agree with this comment and have included additional text commenting on this matter (see lines 346 to 350). In summary, we observed median $\Delta N_{10-50}$ of $12 \times 10^2$, $9.8 \times 10^2$, and $9.6 \times 10^2$ particles $cm^{-3}$ for strong inversion, weak inversion, and unstable atmosphere, respectively. From strong to weak inversion $\Delta N_{10-50}$ decreased by 18%, while this delta remained practically constant between weak inversion and unstable atmosphere. In general, $\Delta N_{10-50}$ exhibited the lower change among the stability conditions, given that from strong inversion to weak inversion, $\Delta N_{50-100}$ reduced by 50 % and $\Delta N_{100-600}$ by 60 %.

We hypothesize that the almost constant $\Delta N_{10-50}$ is explained by the predominant sources of ultrafine particles: secondary aerosol particles, sea salt, and traffic emissions (Leoni et al., 2018). All these three sources might have an impact on the local and background concentrations at the study site. From the hourly profiles of $N_{10-50}$, peaks can be observed during typical commuting hours (6:00-09:00 and 16:00-17:00) at Retje and the background stations (Fig. 1).

[Figure]

**Figure 1: Hourly variation of the $N_{10-50}$ in the village and the background stations.**

- Figures 4 and 5. Captions: Please, remove "black dots" at the end of the caption. Check whiskers: do represent 25-75%?

Response: Thanks for this remark. We have corrected the caption in Fig. 4 by removing "black dots." The whiskers represent the minimum and maximum values without outliers; the distances between the

whiskers contain 25 % of the data at each side of the distribution. The captions in Figures 4 and 5 were updated to avoid misunderstandings.

Note: In Fig. 4, the changes in OA ($\Delta$OA, Fig. 4a) were excluded since, only in this section, the mass of $PM_1$ at the background station was calculated and used to estimate OA. However, the maximum mobile diameter covered by the MPSS in the background (600 nm) does not cover $PM_1$ in its entirety, making it inaccurate to calculate $PM_1$ and, therefore, OA mass.

- Figure 6. I do understand the increase of absolute concentrations during strong inversions. How do you explain the increase of the relative contribution of BrC with respect to BCtotal? Is it because strong inversions are mainly produced at night when domestic heating emission are more important? Or because you assume all BB emissions are local while traffic emissions are also external? Based on the results obtained in the paper, do you believe this source apportionment is realistic? Have you compared with BC SA at the reference site?

Response: We believe the larger fraction of absorption attributed to BrC during strong inversion is a combined result of local emissions and accumulation because of the weather conditions, i.e., during strong inversion, there might be an increase in local wood burning due to lower temperatures, but the accumulation of the local pollution is higher due to reduced vertical mixing and ventilation. The last minimizes the influence of external sources of pollution, and so we hypothesize that the vast fraction of BrC proceeds from wood burning at the study site. On the other hand, during unstable atmosphere, there is a higher chance that local pollution gets diluted and also mixed with external sources such as traffic from national and regional routes or long-range transported aerosols. However, there are traffic emissions at the study site from local households, which might eventually contribute to local measurements, though this contribution is rather small (less than 100 vehicles circulating daily) and concentrated during specific periods of the day (Glojek et al., 2020, 2022).

We do have confidence that the absorption apportionment is representative of the dynamics occurring in the study site. Using a different approach for the same study site in winter, Glojek et al. (2020) found an average contribution of 63 % from biomass burning to eBC mass concentrations, compared to 37 % for traffic emissions.

Respect the last question on this comment, we are unsure about the meaning of SA in the referee's question. We assume this might be scattering albedo and, consequently, we have estimated the SSA from our Mie modeling results. The values of SSA were significantly lower than the usual values reported in the literature. At 470 nm, we found a mean SSA of 0.48 for strong inversion and 0.51 for unstable atmosphere. At 660 nm, SSA were 0.61 and 0.63 for strong inversion and unstable atmosphere. The values of SSA might not be reliable enough to draw significant conclusions. In general, from other studies we know that particles with a comparatively higher fraction of OC (and BrC) than EC, have SSA closer to 1 at 660 nm and close to 0.9 at ~ 400 nm (Pokhrel et al., 2016).

References

Brown, S. G., Lee, T., Roberts, P. T. and Collett, J. L.: Variations in the OM/OC ratio of urban organic aerosol next to a major roadway, J. Air Waste Manag. Assoc., 63(12), 1422–1433, doi:10.1080/10962247.2013.826602, 2013.

Buonanno, G., Dell'Isola, M., Stabile, L. and Viola, A.: Uncertainty budget of the SMPS-APS system in the measurement of PM 1, PM2.5, and PM10, Aerosol Sci. Technol., 43(11), 1130–1141, doi:10.1080/02786820903204078, 2009.

Glojek, K., Gregorič, A., Močnik, G., Cuesta-Mosquera, A., Wiedensohler, A., Drinovec, L. and Ogrin, M.: Hidden black carbon air pollution in hilly rural areas—a case study of Dinaric depression, Eur. J.

Geogr., 11(2), 105–122, doi:10.48088/ejg.k.glo.11.2.105.122, 2020.

Glojek, K., Močnik, G., Alas, H. D. C., Cuesta-Mosquera, A., Drinovec, L., Gregorič, A., Ogrin, M., Weinhold, K., Ježek, I., Müller, T., Rigler, M., Remškar, M., Van Pinxteren, D., Herrmann, H., Ristorini, M., Merkel, M., Markelj, M. and Wiedensohler, A.: The impact of temperature inversions on black carbon and particle mass concentrations in a mountainous area, Atmos. Chem. Phys., 22(8), 5577–5601, doi:10.5194/acp-22-5577-2022, 2022.

Leiva, G. M. A., Araya, M. C., Alvarado, A. M. and Seguel, R. J.: Uncertainty estimation of anions and cations measured by ion chromatography in fine urban ambient particles (PM 2.5), Accredit. Qual. Assur., 17(1), 53–63, doi:10.1007/s00769-011-0844-4, 2012.

Leoni, C., Pokorná, P., Hovorka, J., Masiol, M., Topinka, J., Zhao, Y., Křůmal, K., Cliff, S., Mikuška, P. and Hopke, P. K.: Source apportionment of aerosol particles at a European air pollution hot spot using particle number size distributions and chemical composition, Environ. Pollut., 234, 145–154, doi:10.1016/j.envpol.2017.10.097, 2018.

Pokhrel, R. P., Wagner, N. L., Langridge, J. M., Lack, D. A., Jayarathne, T., Stone, E. A., Stockwell, C. E., Yokelson, R. J. and Murphy, S. M.: Parameterization of single-scattering albedo (SSA) and absorption Ångström exponent (AAE) with EC/OC for aerosol emissions from biomass burning, Atmos. Chem. Phys., 16(15), 9549–9561, doi:10.5194/acp-16-9549-2016, 2016.

Srinivas, B. and Sarin, M. M.: PM2.5, EC and OC in atmospheric outflow from the Indo-Gangetic Plain: Temporal variability and aerosol organic carbon-to-organic mass conversion factor, Sci. Total Environ., 487(1), 196–205, doi:10.1016/j.scitotenv.2014.04.002, 2014.

Xing, L., Fu, T. M., Cao, J. J., Lee, S. C., Wang, G. H., Ho, K. F., Cheng, M. C., You, C. F. and Wang, T. J.: Seasonal and spatial variability of the OM/OC mass ratios and high regional correlation between oxalic acid and zinc in Chinese urban organic aerosols, Atmos. Chem. Phys., 13(8), 4307–4318, doi:10.5194/acp-13-4307-2013, 2013.

---

## Author Comment (AC3)

**Correspondence to Anonymous Referee #3**

The authors extend their appreciation for the constructive feedback received from Anonymous Referee #3. Below, we present the responses to every comment, with the referee's remarks in blue font and the authors' responses in black.

Comments to the manuscript: "Optical properties and simple forcing efficiency of the organic aerosols and black carbon emitted by residential wood burning in rural Central Europe" by Cuesta-Mosquera et al.

In this manuscript the Authors present the results from a winter measurement campaign performed in a rural European site strongly affected by RWB emissions and characterized by strong thermal inversions. The site location and emission characteristics allow for a robust optical characterization of RWB OA. The results from a simple forcing efficiency estimation are also reported.

The manuscript is well written and the results consistently reported. The paper can be published in ACP after some minor revisions reported below.

- 7, line 171: Has the article about the harmonization factor H been published at the time of this review? Can the authors provide some more information? One reference about H (1.76) is Savadkoohi et al., 2023 (https://doi.org/10.1016/j.envint.2023.108081).

Response: Many thanks for the recommendation and the reference to Savadkoohi et al., 2023. The paper discussing the Harmonization factor is still not available. However, we have included a reference to a report from ACTRIS where the harmonization factor is introduced (Müller and Fiebig, 2021, https://www.actris-ecac.eu/particle-light-absorption.html). Furthermore, we have referenced Savadkoohi et al. (2023) as a study case where the AE33 absorption coefficients are harmonized using the H factor from ACTRIS.

Further information about the harmonization factor $H$ has been included in the revised version of the manuscript (see lines 186 to 199).

- In this manuscript the signal at 950 nm is used as reference to calculate eBC, MAC and to separate BC and BrC contribution to absorption in the 370-880 nm spectral range. Normally the 880 nm signal is used for these objectives as a compromise between excluding the absorption from OA and having a good signal-to-noise ratio. By using the 950 nm as reference, automatically a small OA absorption at 880 nm is allowed, whereas OA absorption is usually (in literature) excluded at this wavelength. Can the authors provide some more details about the choice of using the 950 nm?

Response: The decision to use 950 nm as a reference wavelength to calculate eBC mass concentrations, aimed to extend the wavelengths available to calculate optical properties, including the AAE, do BC/BrC apportionment and estimate Simple Forcing Efficiency. Given the large pollution and light absorption measured at Loški Potok, we consider that the signal-to-noise ratio in the near-IR is not an issue in our study.

- It might be more useful to present in figure 3d the first derivative of the potential temperature with horizontal lines highlighting weak, strong, unstable, neutral conditions.

Response: Thanks for this observation. Figure 3d was modified to show the potential temperature gradient.

- Equation 13: Is there any specific reason why an AAE of 1 was used?

Response: We used AAE$_{BC}$ = 1 as an approximation based on the generalized use among the aerosol scientific community. Nevertheless, we understand that AAE$_{BC}$ might range predominantly between ~0.8 to 1.4; this deviation was included in our calculations of uncertainty for the apportioned light absorption coefficients of BC and BrC.

- 17. Lines 379-389: Here the authors present the Angstrom exponent of BrC absorption that was calculated between 370 and 590 nm. Thus, the BrC absorptions calculated at 660 and 880 nm were excluded from the BrC AE calculation. In fact, the authors explain that if the BrC AE is calculated between 370 and 880 nm, then a 50% overestimation of BrC absorption at 370 nm (obtained from equation 14) is observed.

  However, it would be useful if the authors could provide more details about how they "simulated" the BrC absorption at 370 nm using the calculated BrC AE. If I well understand, the "simulated" BrC absorption at 370 nm was calculated from the BrC at 880 nm using the BrC AE from 370 and 880 nm and this "simulated" BrC absorption at 370 nm overestimates by 50% the BrC absorption obtained using equation 14. Consequently, the best simulation of BrC absorption at 370 nm was obtained using the AE from 370 and 590 nm. Thus, the BrC absorption at 370 nm was simulated from the BrC absorption at 590 nm using the AE calculated from 370 and 590.

  Is the procedure described above the one used by the authors?

  It would also be useful if the authors could explain in more detail the reasons why the absorptions at 660 nm and 880 nm were reasonably excluded. The authors report that this could be due to the presence of internally mixed aerosol particles. However, since the procedure described here and used to separate the absorption by BC and BrC is widely used, more details regarding why one needs to go down two wavelengths (from 880 to 590 nm) to calculate the AE should be given.

Response: To determine the light absorption coefficients of BrC in the whole spectrum, we assume that the total absorption corresponds to the contributions of BC and BrC (Eq. 1), and use the mathematical expression describing the AAE (Eq. 2):

$$b_{abs}(\lambda) = b_{abs,BC}(\lambda) + b_{abs,BrC}(\lambda), \tag{1}$$

$$\frac{b_{abs,BC}(\lambda_1)}{b_{abs,BC}(\lambda_2)} = \left(\frac{\lambda_1}{\lambda_2}\right)^{-AAE_{BC}}, \tag{2}$$

To solve the system of equations, we assume that $AAE_{BC} = 1$, and that the total absorption in the near-IR is totally attributed to BC. Therefore, if we take $\lambda_2$ as 950 nm (near-IR), we have that:

$$b_{abs,BC}(950\ nm) = b_{abs}(950\ nm), \tag{3}$$

Now equation 2 can be rearranged as follows,

$$b_{abs,BC}(\lambda_1) = b_{abs}(950\ nm) * \left(\frac{\lambda_1}{950}\right)^{-1}, \tag{4}$$

And $b_{abs,BrC}(\lambda)$ can be expressed as:

$$b_{abs,BrC}(\lambda_1) = b_{abs}(\lambda_1) - b_{abs,BC}(\lambda_1), \tag{5}$$

Where $\lambda_1$ would be any wavelength and the apportioned light absorption can be calculated for the range 370 to 880 nm.

The apportioned BrC light absorption coefficients are fitted through power law in order to calculate AAE$_{BrC}$. For this, we initially used the range of wavelengths covered by the AE33 and obtained AAE$_{BrC,370-880\ nm}$ = 5.5. Nevertheless, the fitting along the whole spectrum produced a significant overestimation of the $b_{abs,BrC}$ at 370 nm, one of the most important wavelengths in our study to report

OA optical properties, given the significant contribution of BrC to shorter wavelengths absorption. Consequently, we estimated AAE$_{BrC}$ for two segregated intervals: 370 to 520 nm and 590 to 880 nm. The slopes from each wavelength range are comparatively different, which is a clear indicator that one single AAE for BrC might not be representative; furthermore, existing studies have demonstrated that AAE$_{BrC}$ is strongly wavelength-dependent (Hoffer et al., 2006; Utry et al., 2014). To improve the representation of the slope change, we have modified Fig. 7a using a log-log scale:

[Figure]

**Fig. 7a: Power law fittings of the BrC absorption spectra in log-log scale**

The value of AAE is an indicator of aerosol chemical composition and is presumably influenced by the aerosol size. The impact of AAE$_{BrC}$ calculated from segregated wavelengths has been studied. For instance, Utry et al. (2014) obtained improved correlations between the particle modes and geometric mean diameters, levoglucosan/total carbon ratio, and OC/EC ratio using an AAE$_{BrC}$ computed for the range of 355 to 532 nm; in contrast, comparatively poorer correlations were obtained when AAE$_{BrC}$ was estimated for the spectral range 266 to 1064 nm. We have this reasoning in the manuscript (see lines 418 to 424).

References

Hoffer, A., Gelencsér, A., Guyon, P., Kiss, G., Schmid, O., Frank, G. P., Artaxo, P. and Andreae, M. O.: Optical properties of humic-like substances (HULIS) in biomass-burning aerosols, Atmos. Chem. Phys., 6(11), 3563–3570, doi:10.5194/acp-6-3563-2006, 2006.

Müller, T. and Fiebig, M.: ACTRIS In Situ Aerosol: Guidelines for Manual QC of AE33 absorption photometer data. [online] Available from: https://www.actris-ecac.eu/particle-light-absorption.html, 2021.

Savadkoohi, M., Pandolfi, M., Reche, C., Niemi, J. V., Mooibroek, D., Titos, G., Green, D. C., Tremper, A. H., Hueglin, C., Liakakou, E., Mihalopoulos, N., Stavroulas, I., Artiñano, B., Coz, E., Alados-Arboledas, L., Beddows, D., Riffault, V., De Brito, J. F., Bastian, S., Baudic, A., Colombi, C., Costabile, F., Chazeau, B., Marchand, N., Gómez-Amo, J. L., Estellés, V., Matos, V., van der Gaag, E., Gille, G., Luoma, K., Manninen, H. E., Norman, M., Silvergren, S., Petit, J. E., Putaud, J. P., Rattigan, O. V., Timonen, H., Tuch, T., Merkel, M., Weinhold, K., Vratolis, S., Vasilescu, J., Favez, O., Harrison, R. M., Laj, P., Wiedensohler, A., Hopke, P. K., Petäjä, T., Alastuey, A. and Querol, X.: The variability of mass concentrations and source apportionment analysis of equivalent black carbon across urban Europe, Environ. Int., 178(June), doi:10.1016/j.envint.2023.108081, 2023.

Utry, N., Ajtai, T., Filep, Á., Pintér, M., Török, Z., Bozóki, Z. and Szabó, G.: Correlations between absorption Angström exponent (AAE) of wintertime ambient urban aerosol and its physical and chemical properties, Atmos. Environ., 91, 52–59, doi:10.1016/j.atmosenv.2014.03.047, 2014.